# Comparing Apple and Pear Phenology and Model Performance: What Seven Decades of Observations Reveal

**Bianca Drepper [1,\*], Anne Gobin [1,2] , Serge Remy [3] and Jos Van Orshoven [1]**

[1] Division Forest Nature and Landscape, University of Leuven, 3001 Leuven, Belgium;
anne.gobin@vito.be (A.G.); jos.vanorshoven@kuleuven.be (J.V.O.)

[2] Vlaamse Instelling voor Technologisch Onderzoek NV, 2400 Mol, Belgium

[3] Proefcentrum fruitteelt vzw, 3800 Sint-Truiden (Kerkom), Belgium; serge.remy@pcfruit.be

\* Correspondence: bianca.drepper@kuleuven.be; Tel.: +32-16-374605

**Abstract:** Based on observations for the beginning of the flowering stage of *Malus domestica* (apple) and *Pyrus communis* (pear) for the 1950–2018 period, phenological trends in north-eastern Belgium were investigated in function of temperatures during dormancy. Moreover, two different phenological models were adapted and evaluated. Median flowering dates of apple were on average 9.5 days earlier following warm dormancy periods, and 11.5 days for pear, but the relationship between bloom date and temperature was found not to be linear, suggesting delayed fulfilment of dormancy requirements due to increased temperatures during the chilling period. After warm chilling periods, an average delay of 5.0 and 10.6 days in the occurrence date of dormancy break was predicted by the phenological models while the PLSR reveals mixed signals regarding the beginning of flowering. Our results suggest overlapping chilling and forcing processes in a transition phase. Regarding the beginning of flowering, a dynamic chill model coupled to a growing degree days estimation yielded significantly lower prediction errors (on average 5.0 days) than a continuous chill-forcing model (6.0 days), at 99% confidence level. Model performance was sensitive to the applied parametrization method and limitations for the application of both models outside the past temperature ranges became apparent.

**Keywords:** climate change; pome fruit; apple; pear; sequential modelling; dynamic modelling; dormancy break; chilling requirements; partial least square regression

## 1. Introduction

In 2018, pome fruit orchards represented 2.4% of the agricultural land in the Northern Belgian region Flanders (~15,000 hectares), of which 74% were attributed to the one apple cultivar 'Jonagold' (*Malus domestica)* and one pear cultivar 'Conference' (*Pyrus communis*) [1]. Belgium's production of 'Conference' is about 35% of the total European production in its favorable Atlantic climate. However, climate change might alter these conditions, and during recent years, the Belgian fruit sector has more frequently experienced weather conditions that have heavily compromised the quantity and quality of fruits. A severe frost night in 2017 (19–20 April), that coincided with the end of pear bloom and beginning of bloom of most commercially exploited apple cultivars, had in Belgium the Europe-wide highest losses: 68% yield loss on average for apple and 7% loss for pear as compared to the mean of the nine preceding years [2]. The difference between apple and pear losses suggests that the impact of extreme events or adverse weather conditions relates to the timing of phenological stages and the differences therein.

Other weather-related hazards were several light frost nights, frequent local hailstorms, and three heatwaves in 2019, an extensive drought period for more than two months in combination with a heat wave in 2018, water excess and storms in 2016, and a devastating summer storm in 2011. The consequences of extreme weather in addition to the Russian embargo on fruit imports since 2014 led the entire Belgian fruit sector into an officially recognized crisis, to the extent that federal and regional governments are preparing policies to increase resilience in the face of more frequent extreme weather events and in the context of a changing climate in general.

Therefore, a better understanding is needed of the impacts of warming on phenology in the specific case of apple and pear cultivation in Flanders. By means of adapted phenological models, variations in flowering and the related frost-sensitive stage can be estimated across temporal and spatial scales, which can support decision-making regarding (new) orchard locations and cultivar selection.

Following [3], the part of the growth cycle between late summer and budburst is conceptualized as dormancy period, and can be decomposed in a sequence of paradormancy, endodormancy, and ecodormancy phases. Endodormancy, i.e., 'chilling period', is induced by lower temperatures in late fall. The moment that a species' specific exposure to cold temperatures has been fulfilled, ecodormancy is induced and the accumulation of growing degree hours becomes the determining process for further phenological development [4], i.e., 'forcing period'. From then onwards, warmer temperatures favor earlier and no longer later bloom. Insolation and temperature commonly explain substantial shares of the inter-annual variability of bloom dates, but currently no model based on chilling and temperature forcing has been published that yields satisfying accuracy to predict bloom dates [5]. Furthermore, the existing phenology models have been developed for certain growing areas (like Israel [6], Finland [7], Germany [8]) but their temporal and geographical transferability is very limited [9], making the parametrization of a phenological model to local bloom observations necessary. In general, limitations of phenological models are related to two factors: The difficulty to observe the transition between dormancy phases and the low number of available observations as compared to the number of variables involved. Statistical methods like partial least square regression (PLSR, of air temperatures and phenology), have been used to overcome the challenge of the limited number of observations (bloom dates) with regard to the number of variables (temperature of individual days) [10].

The evolution of the timing of the bloom is decisive for selecting adequate and 'climate proof' cultivars with regard to frost risk. As suggested by an Austrian study [11], there is a tendency towards an earlier timing of the last spring frost, but late occurrences are not excluded since inter-annual variability is substantial and regionally blocked atmospheric conditions are more frequent [12]. For mountainous areas, it might be reasonable to assume chilling requirements are not compromised [11], but for orchards in temperate lowlands, the trajectory is less certain. Studies of phenological changes across Europe reveal a large spatial variability with a consistent tendency towards earlier bloom in higher latitudes and a mixed signal in northern Mediterranean settings. Observations in Nîmes, France (lat. 43.8° N) hint at a trend reversal where the years between 2003 and 2013 show, again, a later bloom than in the 1990s [13]. Projections for future years show similar contrasts: For Styria (Austria, lat. 47.3° N), mountainous terrain, consistently earlier bloom is projected [11], whereas others projected a consistent delay of bloom for northern Spain (lat. 42.4° N) [14]. Observations in the Southern Mediterranean region, Tunisia in particular, showed important delays and even frequent failures of flowering for the case of pistachios [15]. These delays observed in the South are explained by a delayed completion of chilling requirements due to warmer temperatures during the endodormancy phase, whereas the trend towards earlier flowering in the North is mainly explained by a more rapid accumulation of forcing or degree hours as a direct consequence of the increased temperature during the forcing phase [16,17].

Recent experimental studies on cherry trees in Germany [18] suggest that faster accumulation of forcing can substitute up to 50% of lower accumulated chilling, but a minimum amount of chilling remains to be fulfilled before forcing can be effective. This motivates the choice of a sequential modelling approach that explicitly accounts for a cultivar-specific relationship between chilling and forcing

requirements. One such model setup emerged as the most adequate for climate change impact studies from an extensive evaluation of mechanistic models for the beginning of apple blossom in Germany [8] and has been applied in several studies [11,19,20]. Over time, first chilling and consequently forcing accumulate or stagnate, but never reverse. This continuous, undynamic approach has the advantage of fast and transparent computations that allow for easy adaptation to different fruits and local conditions. However, it results in less reliable estimations of chilling requirements [21]. Hence, the continuous model only leads to reasonable results within a narrow range of temperatures, where chilling is fulfilled before an accumulation of growing degree hours is possible.

The chilling and subsequent forcing expressed in this continuous model are thought to hold for the past in Belgium but seem inadequate in a warming world. Reviews of chill and phenology models [5,9,22] suggest a more complex dynamic process [6,23], that accounts more realistically for chilling and forcing processes. We hypothesized that dynamic phenological modelling captures the chilling and forcing requirements in a better manner and can be used to simulate phenological development of apple and pear in a warming climate.

The major objectives of this paper are (1) to analyze phenological responses to increasing temperatures during chilling and forcing periods, with the hypothesis that bloom dates are not proportionally advancing; (2) to detect shifts in the beginning of the forcing phase using partial least square regression and to compare these shifts with modelled dormancy break dates; (3) to quantify chilling and forcing requirements for a range of cultivars of apple and pear at a study site in Belgium, while comparing the selected continuous and dynamic + growing degree hour (GDH) phenological models and their sensitivity to parameter optimization.

## 2. Materials and Methods

### 2.1. Study Area

In Flanders, the northern administrative region of Belgium, the fruit growing area is mostly concentrated on a plateau with low rolling hills and elevations between 30 and 200 m above sea level, divided over the natural regions Hageland and Haspengouw. A more recent cluster is found in the north (west of Antwerp) in the Waasland natural region (Figure 1a). The study site is located in the first zone, where the annual average temperature during the period 1950–2018 was 10.1 °C.

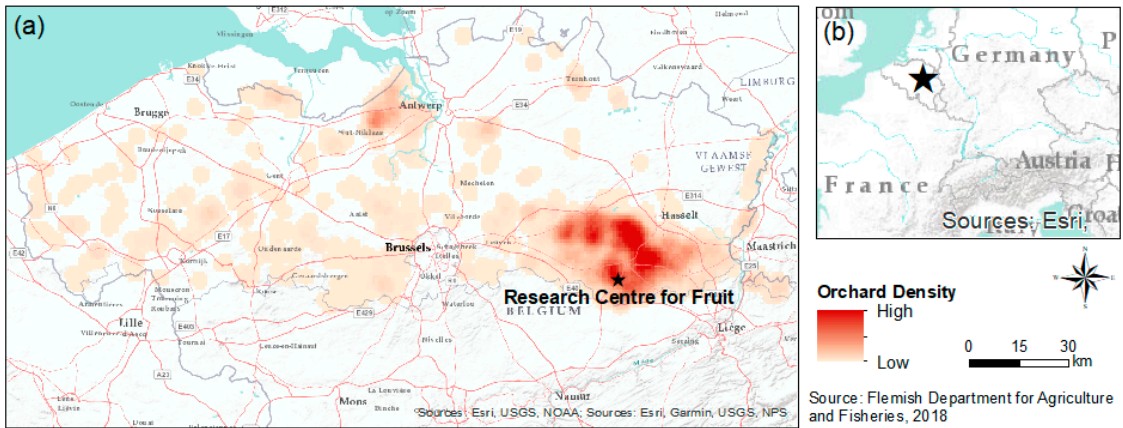

**Figure 1.** Distribution of apple and pear orchards in the Flemish Region, Belgium (**a**), with locator map (**b**). The star marks the study site, the Research Centre for fruit (50°46′ N, 5°09′ E, 92 m above sea level).

### 2.2. Phenology and Meteorology

The timing of the phenological stages according to the definitions of the Biologische Bundesanstalt, Bundessortenamt, and CHemical industry (BBCH) [24] has been recorded since 1950 for the main apple and pear cultivars by the Research Centre for Fruit (Pcfruit vzw), Sint-Truiden (Kerkom), Belgium.

For 19 cultivars of economic relevance, observations for the beginning of flowering (BBCH61) were transcribed in machine-readable format and inconsistent observations were deleted. Based on the length and completeness of the records, four pear varieties ('Conference', 'Durondeau', 'Doyenné', and 'Triomphe') and four apple varieties ('Jonagold', 'Golden Delicous', 'Boskoop', and 'Cox Orange') were retained.

Time series of daily minimum and maximum temperatures were recorded by a synoptic weather station since 1950 at the research center with only three missing records (0.01%), which were gap filled using the average of the preceding and following day. Hourly data were computed as an idealized daily temperature curve that uses a sine function for daytime warming (sunrise to sunset) and a logarithmic decay function for night time cooling [25], as implemented in the function *'make_hourly_temps'* from the R package 'chillR' [26]. The decay functions were found to compare well with the observed hourly temperatures for exemplary years 2016 and 2017 from an identical synoptic weather station at 1 km from the station of the long-term records. The monitored orchards were within 500 m distance from this synoptic weather station. All analyses were done using Python programming language version 3.6, making use of the rpy2 translator [27] to call scripts published in R language [26].

## 2.3. Ranking Observations to Derive Past Change Signals

Climate change signals were studied based on 68 years of observations (1951–2018) for which temperatures for the full dormancy period were available (bloom observation for 1950 had to be excluded for this reason). A study of climate impact on growth and productivity of arable crops in Belgium showed significant differences in temperature before and after a change point in 1988 [28], which allowed for the comparison of weather impacts on the growing season and on sensitive phenological stages for two contrasting climatic periods. Change point analysis was also applied directly to phenology data for West Germany, Switzerland, and the Balkans [29–31] and suggested break points around 1988.

However, for different temperate locations, 2-segmented pairwise constant models, with such a breakpoint and used to correlate temperature and phenological stages, did not perform consistently better than linear models (according to the modified Bayesian information criterion) [13]. This suggests that complex processes such as phenological development need further analysis to disentangle effects of warming on chilling and forcing. Therefore, contrasting periods were selected based on rankings of mean temperatures during the full dormancy periods (October to April inclusive), chilling periods (October, November, and December), and forcing periods (January to April inclusive). The inclusion of January to the forcing period in contrast to [13] is motivated based on the results of the PLSR.

Since the dormancy state overlaps two calendar years, observations were attributed to the year of flowering (i.e., October 2017 to April 2018 were associated to 2018). Contrasting periods were selected in two ways: By dividing the ranked 68-year period (i) in two halves of 34 years (referred to as 'cold' and 'mild'); and (ii) in three parts (the 25% cold years ($n = 17$); the mild 25% years ($n = 17$); and the remaining average 50% ($n = 34$). Segments were based on the number of years, not the values themselves, with no special treatment for extremes.

Considering only hourly temperatures in October, November, and December (OND), relevant hours were counted using different thresholds for effective chilling temperatures as [0, 10 °C] (continuous model), [−2, 14 °C] (dynamic + GDH model), and [3, 9 °C] (the most effective temperatures for chilling [23]). This was done using the contrasting segments of years in terms of average temperature during OND and, for comparison, the effective hours were expressed as a percentage of the total number of hours in OND.

## 2.4. Partial Least Square Regression

Partial least square regression (PLSR) is a method that uses a linear multivariate model to relate two data matrices **X** and **Y** to each other. While it is commonly used in chemometrics [32], its application to plant phenology is novel, though it has generated good results in studies on walnut in

California [10,33], cherry in Germany [33], and apple in China [34]. The method goes beyond common multiple linear regression by modelling the structure of the matrices **X** and **Y**, thereby allowing to link a comparatively small number of observations for the dependent variables, i.e., date of achieving phenological stages, to a large dataset of highly auto-correlated variables, i.e., meteorological time series. In this application, the observed flowering date per year (Julian day of year (DOY)) was related to the antecedent temperatures [°C] on a daily basis. The beginning of the dormancy period has been approximated as 1 October, after finding the sign change analysis not to be very sensitive to the starting date. This way, model artefacts, i.e., chilling contribution in August, were reduced.

Daily temperature, rather than modelled daily chilling and forcing accumulation, was chosen as independent variable to enable the comparison of the model-independent output with both selected phenological models.

In PLSR, a variable compression is performed on the covariance matrix to reduce auto-correlation between predictor variables and to keep only linear combinations of maximum covariance between the explanatory variables **X** (daily temperatures) and the dependent variables **Y** (yearly bloom date) [10]. Temperatures of each day were scaled by their standard deviations, so that all predictors had the same variation. This allowed to assess the influence of variation through time of variables independent of their magnitude. Following [10], an 11-day running mean filter was consequently applied on the daily temperature data for noise reduction.

The function yields two metrics for each day: The variable importance in the projection (VIP) and the regression coefficient. The VIP indicates the weighted sum of squares of the PLS loadings and represents the relative importance of the given predictor (x-variable) in the PLS model. VIP values above a fixed threshold value of 0.8 are considered significant [35]. The sign of the regression coefficient indicates the direction of the relationship between an x-variable and the y-variable, while the magnitude of the regression coefficient reflects the VIP and is proportional to the contribution of warmer temperatures to the timing of blossom. In other words, positive coefficients suggest a delaying effect and negative ones an advancing effect of the 11-day average daily mean temperature on bloom date for the given time period, while higher absolute values indicate the strength of the effect in a given period of the year.

The function 'PLS_pheno' from the 'chillR' R-package [26] was used on the temperature records (68 years) to delimit the chilling and forcing periods per cultivar; as well as for two contrasting segments within the time series: The 34 cold and 34 mild years. Using the tabular output of the PLS function, the date of the beginning of forcing is inferred as the first occurrence of a sign change from a positive to a negative coefficient, followed by (i) at least three days out of 4, 7, or 17 days (iterative process) with the occurrence of a VIP value >0.8 and a negative coefficient, and (ii) a median negative coefficient for the following 10 days. Only dates after 1 December were considered.

### 2.5. Phenological Models

Two conceptually different models of the chilling and forcing processes towards flowering were calibrated and validated for each considered cultivar. The calibration years consisted of the even years between 1950 and 2018 while uneven years were used to validate the models, similarly to [8]. Thereby both the calibration and validation datasets included cold and mild years.

### 2.5.1. Continuous Model

The model referred to as 'continuous model' (as opposed to dynamic) largely draws on the 'sequential model' or 'M2', [7,36] as described in the phenological model comparisons for fruits in Germany [8], with a few modifications as given in the equations below.

Per year with bloom observations, antecedent chilling and forcing was estimated on a daily basis, starting from a fixed day, 1 August. Chilling units ($R_c$) were accumulated (Equations (1)–(3)) until the accumulated chilling ($S_c$) equaled the chilling requirement C (Equation (4)). Only positive temperatures

were taken into account as this was considered most appropriate for the German context [8] and, by extension, also for Belgium. The day $t_1$ is considered the modelled day of dormancy break.

$$R_c(T_i) = 0 \quad if \ T_i \leq 0.0 \ or \ T_i > T_{Up} \tag{1}$$

$$R_c(T_i) = \frac{T_i}{T_{Bc}} \quad if \ 0.0 < T_i \leq T_{Bc} \tag{2}$$

$$R_c(T_i) = \frac{T_i - 10.0}{T_{Bc} - 10.0} \quad if \ T_{Bc} < T_i \leq T_{Up} \tag{3}$$

$$S_c(t) = \sum_{i=t_0}^{t} R_c(T_i), \ where \ S_c(t_1) =: C \tag{4}$$

where $R_c$ expresses daily chilling units that accumulate to $S_c$; $T_{Bc}$ and $T_{Up}$ are variable parameters that determine temperature ranges with defined chilling efficiency; and $T_i$ is the average temperature of the day $t_i$.

Following [37], a cultivar-specific exponential relationship between the chilling and the forcing requirement is assumed following Equation (5).

$$F = a * \exp(bS_c(t_1)) \tag{5}$$

where the factors $a$ and $b$ determine the dependent forcing requirement $F$ in function of the calibrated chilling requirement $C$. The forcing temperature ($R_f$) was accumulated according to Equations (6) and (7).

$$R_f(T_i) = 0, \ if \ T_i \leq T_{Bf} \tag{6}$$

$$R_f(T_i) = \frac{28.4}{(1 + \exp(-0.185(T_i - T_{Bf} - 18.4))}, \ if \ T_i > T_{Bf} \tag{7}$$

where $T_{Bf}$ is a similar parameter to $T_{Bc}$ to adjust forcing efficiency ranges, starting from day $t_1$, until day $t_2$, when the accumulated forcing ($S_f$) equals the forcing requirement $F$, as expressed by Equation (8):

$$S_f(t) = \sum_{i=t_i}^{t} R_f(T_i), \ where \ S_f(t_2) := F. \tag{8}$$

In summary, six parameters (Table 1) needed to be adjusted for each cultivar within ranges based on values for different fruit growing regions in Germany [8,20], where annual mean temperature between 1961 and 2005, ranged from 8.7 °C to 10.1 °C. To adapt the model to our particular study site, $T_{Bf}$ and $T_{Bc}$ were subjected to calibration.

**Table 1.** Parameter space—in squared brackets: Lower and upper bound and step; in round brackets: Initial guess (best fit parameters and boundaries based on parametrizations for German fruit growing regions [20]).

|  | **Equation** | **Apple** | **Pear** |
|---|---|---|---|
| $T_{Bf}$ [°C] | Equation (6) | [4, 6, 0.1] (4.5) | [2, 5, 0.1] (3.8) |
| $T_{Bc}$ [°C] | Equations (1) and (3) | [2, 10, 0.1] (3.5) | [2, 10, 0.1] (5.2) |
| $C$ [chill units] | Equation (4) | [40, 70, 1] (50) | [30, 80, 1] (50) |
| $a$ | Equation (5) | [140, 200, 1] (172) | [200, 220, 1] (206) |
| $b$ | Equation (5) | [−0.05, −0.001, 0.001] (−0.001) | [−0.05, −0.01, 0.001] (−0.002) |
| $T_{Up}$ [°C] | Equations (1) and (3) | [8, 14, 0.1] (10.4) | [8, 14, 0.1] (10) |

## 2.5.2. Dynamic + GDH Model

The selected dynamic chill model was originally developed to study dormancy break of peach trees in Israel [6,23] and is, to date, widely accepted as the most robust phenological model for this

purpose [5,9,21]. As it does not predict phenological stages beyond dormancy break, it has been combined with a growing degree hour (GDH) accumulation estimation following [4], using the function 'Bloom_prediction3' in the R package 'chillR' [26]. In this study, the term 'dynamic + GDH model' always refers to this combined setup.

The function takes only two parameters: A cultivar-specific chilling requirement (CR) in the unit 'Chill Portions' of the base model and a heat requirement (HR) in GDH. Dormancy break is reached when the accumulated daily chill portions equal the given CR. The day of flowering was determined as the day of year when the GDH equaled the HR, not considering accumulation prior to dormancy break since the inherent assumption is that GDH accumulation only starts when rest completion is reached. Unlike the continuous model, chilling and forcing requirements were given independently from each other and substitution, as documented for other tree species, is not considered in the coupling of chilling and forcing modules [38].

Chilling accumulation efficiency followed a bell-shaped curve with a maximum efficiency at 6 °C and zero effect below −2 °C and above 14 °C. Higher temperatures can undo chill portions in a dynamic way. For the detailed equations, we refer to the original documentation of the model [23,26]. The start of accumulation was defined in the function as 1 November. Daily GDH were derived using Equations (9) and (10).

$$GDH = \frac{T_U - T_B}{2} * \left(1 + \cos\left(\pi + \pi * \frac{(T_i - T_B)}{(T_u - T_B)}\right)\right), \; if \; T_B < T_i < T_U \tag{9}$$

$$GDH = (T_U - T_B) * \left(1 + \cos\left(\frac{\pi}{2} + \frac{\pi}{2} * \frac{(T_i - T_U)}{(T_C - T_U)}\right)\right), \; if \; T_U < T_i < T_C \tag{10}$$

where the base temperature $T_B$ is 4 °C, the optimum temperature $T_U$ is 25 °C, and the critical temperature $T_C$ is 36 °C, following predefined thresholds for fruit trees [4].

To calibrate the model parameters for Belgian conditions, the parameters were optimized using a range of possible parameters adapted from [26], as shown in Table 2.

**Table 2.** Parameter space in squared brackets: Lower and upper bound and step; in round brackets: Initial guess (values approximated based on [26]).

|  | Apple | Pear |
|---|---|---|
| CR | [50.0, 80.0, 1] (60) | [30, 70, 1] (45) |
| HR | [3000, 5000, 10] (4000) | [3000, 4500, 10] (3500) |

### 2.5.3. Parameter Optimization

Both the continuous and the dynamic + GDH model were calibrated to match the observations by cultivar using the SPOTPY package in python [39] and a selection of optimization algorithms: Monte Carlo (MC), Markov chain Monte Carlo, (MCMC), differential evolution Markov Chain (z = with matrix extension [40], DE-MCz), maximum likelihood estimation (MLE), simulated annealing (SA), Latin hypercube sampling (LHS). The root means square error (RMSE) was minimized as objective function in all cases except for the MCMC and DE-MCz algorithms, which come with their own likelihood function (logarithmic probability density).

The parameter distribution was assumed to be uniform for the required parameters, resulting in spaces as described in Tables 1 and 2. The 'best guess' gives the starting point for the optimisation procedure. The step size was adapted to be one order of magnitude below the chosen values. Only the algorithm MCMC is sensitive to the step size of the parameter function. The number of repetitions needed to find a reliable result was calculated with Equation (11) [39] except for the LHS algorithm for which Equation (12) is applicable

$$N = \left(1 + 4 M^2 (1 + (k - d)d)\right)k \tag{11}$$

$$N_{max} = (p!)k - 1 \tag{12}$$

where $N$ = needed parameter iterations, $k$ = the number of parameters of the model, $M$ = inference factor (SPOTPY default $M = 4$), and $d$ = frequency step (the minimum being 0.01), $p$ = the divisions per parameter.

The minimum necessary number of iterations was 390 for the continuous model and 160 for the dynamic + GDH model. Finally, 500 or 200 iterations were called. Both models were evaluated based on the root means square error (RMSE), the mean absolute error (MAE), and the ratio of performance to interquartile distance (RPIQ) [41], which is defined as the ratio of the range between the first and the third quartile of the observations to the RMSE of the prediction, as in Equation (13). Thereby, no assumptions are made about the distribution of the observed values (in contrast to the commonly used ratio of performance to deviation (RPD).

$$RPIQ = \frac{(Q3 - Q1)}{RMSE} \tag{13}$$

where $Q1$ is the first and $Q3$ the third quartile; RMSE is the root mean square error.

## 3. Results

### 3.1. Past Change Signals

All distinct temporal segments within a panel (cold, medium, and mild) in Figure 2 are significantly different from each other at the 99% confidence level. The majority of mild dormancy periods occurred in the second half of the time series (after 1984) (Figure 2a–c), which is four years prior to the aforementioned change point of 1988. This way of dividing the years allowed for disentangling physiological differences between mild and cold winters. Considering the full dormancy period (Figure 2d), the 25% mild years even occurred only in the last three decades, with the exception of 1961. Considering temperatures during the forcing period, Figure 2f revealed that cold springs still occurred due to interannual variability despite the general warming of the air temperature, suggesting that a risk for late spring frost remains.

When the same segments of years were plotted against the corresponding flowering dates, two important observations can be made. Firstly, relating to the upper row of Figure 3: Although all segments were significantly different in terms of daily average temperatures, the corresponding bloom dates were only significantly different (99% confidence level) when the full dormancy period (Figure 3a) or the forcing period (Figure 3c) was considered. For apple vs. pear, resulting advances of the median flowering date were 9.5 vs. 11.25 days (average over the cultivars; 12.5 days for 'Conference', Figure 3a), and 10.3 vs. 13.38 days (14.5 days for 'Conference', Figure 3c). In contrast, with 0.5 days delay and 1.6 days advance (median), no significant change was detected when temperatures during October, November, and December determined the segmentation, Figure 3b. This could imply that rising temperatures during these months (OND) were affecting the bloom date less than the rising temperatures from January to April or that there were antagonistic processes involved. The difference between the cold and mild segments was slightly higher during forcing (2.35 °C, January to April) as compared to during chilling (1.71 °C, October, November, and December).

When looking at the median days of flowering, a nonlinear relationship between the bloom date and the temperature during the full dormancy period is suggested for the cold 25% of all bloom dates (Figure 3d). Blooming occurred significantly later during the medium years (at 99% conf. level) than during the cold 25% years: On average, 13.0 days (apple) and 13.6 days (pear). The difference between the mild 25% years and the medium 50% years was 2.75 days for apple and 1.0 days for pear; none of these values were statistically significant. However, the difference in mean temperature between the first and second (1.36 °C); and second and third segments is nearly identical (1.37 °C). In most cases, the trends were stronger for pear than for apple.

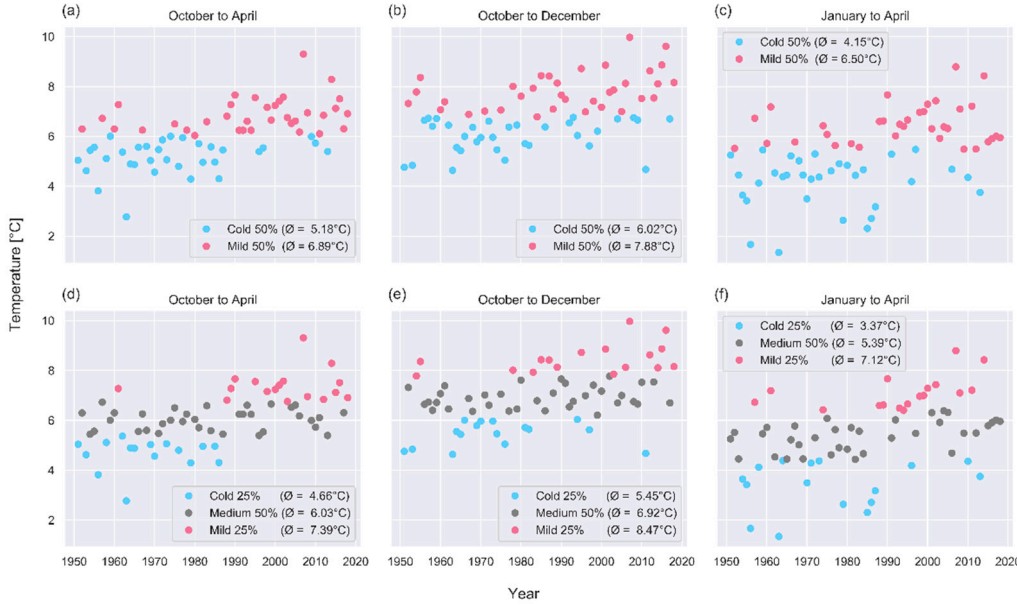

**Figure 2.** Average daily temperatures during the 'full dormancy' period (panels (**a**,**d**)), the approximate chilling period (panels (**b**,**e**)), and forcing periods (panels (**c**,**f**)) studied either after segmentation in halves (34 years each, upper row) or in three parts (17–34–17 years each, lower row). Average temperature refers to daily average air temperature measured at 2 m height during the indicated months at the study site.

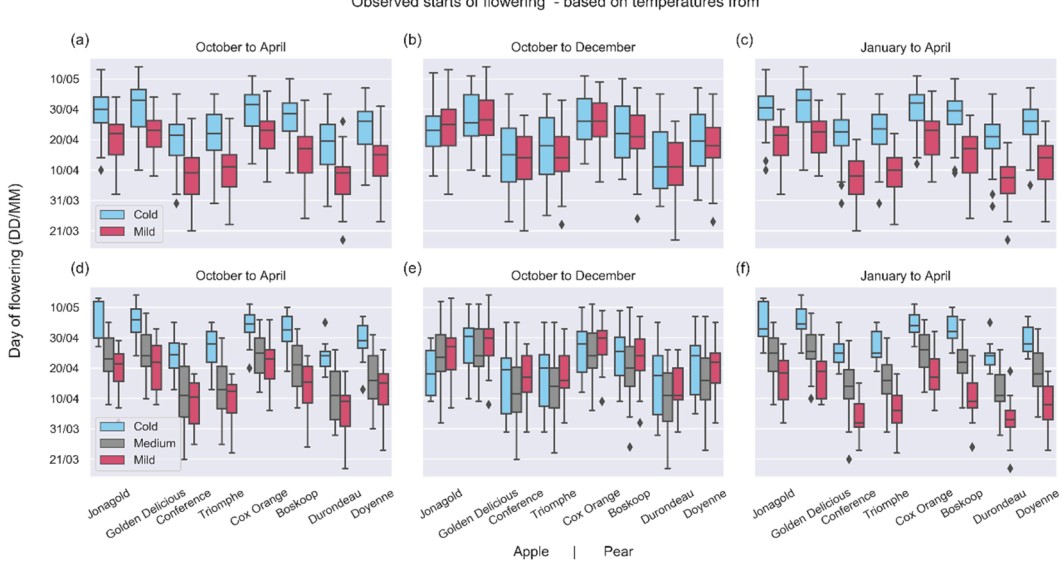

**Figure 3.** Observed bloom date (DD/MM) of eight cultivars during the 'full dormancy' period (panels **a**,**d**), the approximate chilling (panels **b**,**e**), and forcing periods (panels **c**,**f**) presented after segmentation in two halves (34 years each, upper row) or three parts (17–34–17 years each, lower row). Diamonds depict outliers, defined as outside 1.5 times the interquartile range.

The decomposition of the analysis into temperatures during chilling and forcing also revealed disparate patterns: The relationship of rising temperatures during January to April and bloom date was consistently and strongly negative (17.5 days advance from the cold to the medium and 20.75 from the medium to the warm segment). However, the increasing temperature in October, November, and December—Figure 3e—revealed a negative relationship considering the first two segments, and a

positive relationship considering the last two segments. In the case of 'Jonagold', there is even an overall positive relation discernible, which is probably caused by a smaller number of available observations.

Rising temperatures during October, November, and December have consequences on the availability of effective chilling, which was confined between boundaries that depended on the applied model. The share of effective hours was reduced by 0.95–2.58% if the years were split in two halves, with the more pronounced reduction in the most efficient interval (3–9 °C); and by 2.39–4.31% when the cold and mild 25% of the years were considered (Table 3). The strongest signals were found between the mild 25% and the average years when the interval around the most efficient temperature for chilling was considered.

**Table 3.** Percentage of the total number of hours of effective chill hours during the chilling phases (October, November, and December) of sets of cold or mild years as defined by the average temperature during the same months; efficient temperature interval boundaries are given in °C.

| Interval | 50% Cold Chilling Phases | 50% Mild Chilling Phases | 25% Cold Chilling Phases | 50% Average Chilling Phases | 25% Mild Chilling Phases |
|---|---|---|---|---|---|
| [0, 12] | 73.76 | 70.31 | 73.47 | 72.67 | 69.46 |
| [−2, 14] | 85.21 | 83.19 | 85.85 | 85.43 | 83.47 |
| [3, 9] | 49.69 | 46.29 | 42.05 | 40.91 | 37.76 |

Phenological Response to Warming: Partial Least Square Regression

The absolute value of the model coefficients resulting from the PLSR indicate the relative importance of daily temperature on the onset of the bloom period (BBCH61) whereas positive values indicate that an increase in temperature at a given day increased the day of bloom expressed as Julian day of year (DOY), where a higher DOY means a later bloom period. The persistent change from a positive to a negative coefficient is interpreted as a change in dormancy phases, from endodormancy to ecodormancy, further referred to as the beginning of the forcing period.

The PLSR model coefficients for all studied cultivars over all 68 years are shown in Figure 4. For all cultivars, October and November featured mostly positive coefficients, while February, March, and April show only negative coefficients. In all cases, the latter two months have higher relative importance on the bloom date with the highest (absolute) coefficients. December and January show both positive and negative relationships between rising temperatures and bloom date, which can imply that the timings of chilling and forcing overlap.

Focusing on the commercially most relevant pear cultivar in Belgium, 'Conference', a side-by-side comparison of the results of the PLSR for cold and mild years is given in Figure 5, showing that different warming effects can be noted, either by the average temperature during the full dormancy period (ONDJFMA), chilling (OND), or forcing (JFMA). A common element is that on the right side of the figure (mild years), the delaying effect during October, November, and December is notably higher. Also, on the left side (colder years), the advancing effect during February, March, and April is less sensitive to the way the observations have been separated in colder and warmer periods.

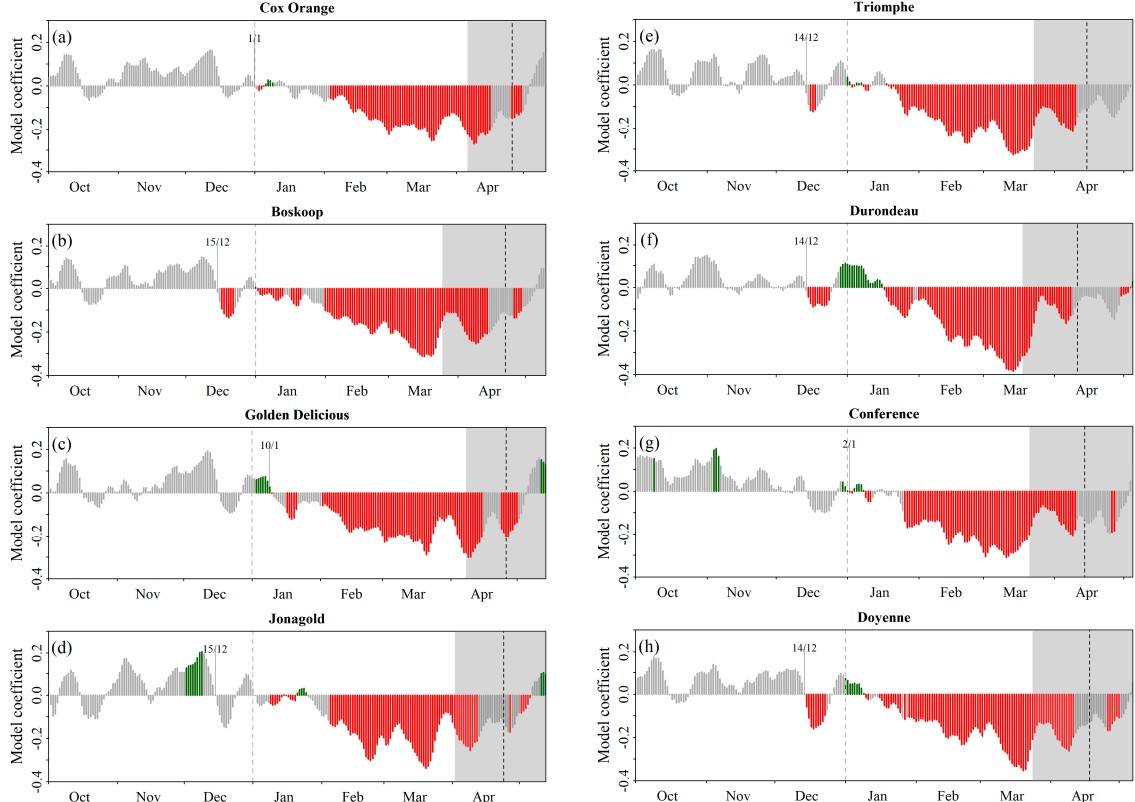

**Figure 4.** Results of the partial least square regression (PLSR) for selected cultivars of apple (**a–d**) and pear (**e–h**) during 68 years between the start date of flowering (BBCH61) and daily average temperatures during the dormancy. Green (positive) and red (negative) areas depict daily coefficients of the model (11-days running average daily mean temperature values) where the variable importance of the projection (VIP) value exceeds the threshold of 0.8, indicating that the corresponding variable was important for the model. In this figure, all years for which bloom date observations were available have been considered. The shaded area represents the range of observed bloom dates, the black dashed line within the shaded area the median value of the bloom date, and the grey dashed line the 1 January as reference. Labelled dates indicate the beginning of the forcing period.

Figure 5a,c resemble each other more than Figure 5b, just as Figure 5d,f resemble each other more than Figure 5e, implying that the warming trend during January to April dominated the ranking, similar to Figure 3. When years were segmented based on *chilling* periods (Figure 5e vs. Figure 5b), the coefficients are more homogenous in the mild segment, where the characteristic transition from positive to negative coefficients was more pronounced. In these mild years, mid-December days have delaying properties (Figure 5e), while the same days were strongly advancing in cold years, Figure 5b. This suggests a shift of the dormancy completion towards a later day, as expected from less chilling temperatures.

When years were segmented such that the *forcing* periods were mild (Figure 5f vs. Figure 5c), a strongly alternating pattern in December and January becomes more pronounced and warming effects are less clearly associated to either delaying or advancing bloom. Daily coefficients are more accentuated in the segment of mild years than in that of cold years.

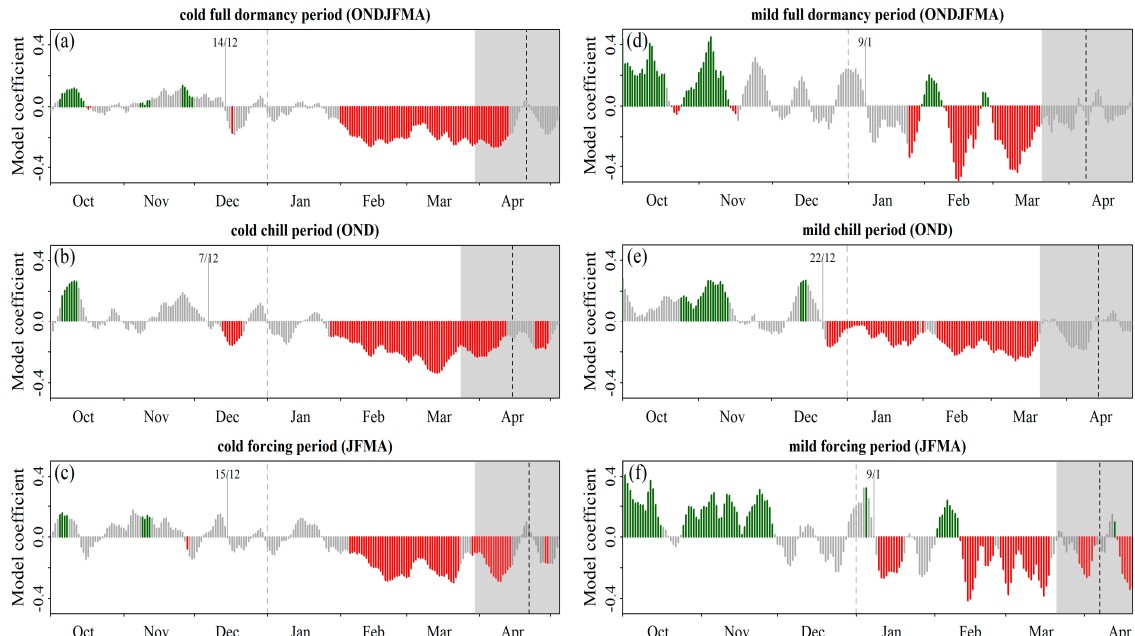

**Figure 5.** Results of the PLSR analysis for 'Conference' pear for the start date of flowering (BBCH61). Green and red areas depict coefficients of predictors (11-day running mean average daily temperature values) where the variable importance of the projection (VIP) value exceeds the threshold of 0.8, indicating that the corresponding variable is important for the model. Years have been separated in cold (left) and mild (right) halves based on average temperature during either the full dormancy phase (October to April (**a**,**d**)); the chilling phase (October to December, (**b**,**e**)); or the forcing phase (January to April, (**c**,**f**)). The shaded area represents the range of observed bloom dates and the black dashed line within the shaded area its median value. The grey dashed line represents the first of January as reference. Labelled dates indicate the beginning of the forcing period.

### 3.2. Shifts of Phenological Phases

From the PLSR results, the beginning of the forcing period can be identified from the sign change of the daily coefficients of the predictors. The earliest occurrence of significant negative coefficients has been shifted towards later dates by at least 15 days in all three rows in Figure 5. Using the average temperatures during the chilling period (October, November, and December) as a criterion to separate cold and mild years, the days of dormancy break simulated by the continuous and the dynamic + GDH models were plotted against the beginning of the forcing periods shown by PLSR in Figure 6. The selected models are shown in Tables A1 and A2.

Different patterns emerge from both models (regarding the mean DOY of dormancy break): under mild temperatures in Figure 6c, compared to Figure 6b, an average 10.6 days later dormancy break was modelled by the continuous model and an average of 5.0 days by the dynamic + GDH model. In all cases but 'Cox Orange', 'Triomphe', and 'Boskoop', the dates simulated by the continuous model were found to be later. For both models, the modelled dates were consistently later (23.3 to 31.25 days) than the dates corresponding to the PLSR sign change which indicate the beginning of forcing. Deviation from the PLSR results were larger for the continuous model than for the dynamic + GDH model. The latter also has a larger standard deviation resulting from one outlier ('Cox Orange') but otherwise a smaller spread. Both models predicted as expected cases where the dormancy break approximated the beginning of forcing, especially for pear but also 'Golden Delicious' after cold chilling periods, but the dynamic + GDH model does so more strongly after milder chill periods (Figure 6c).

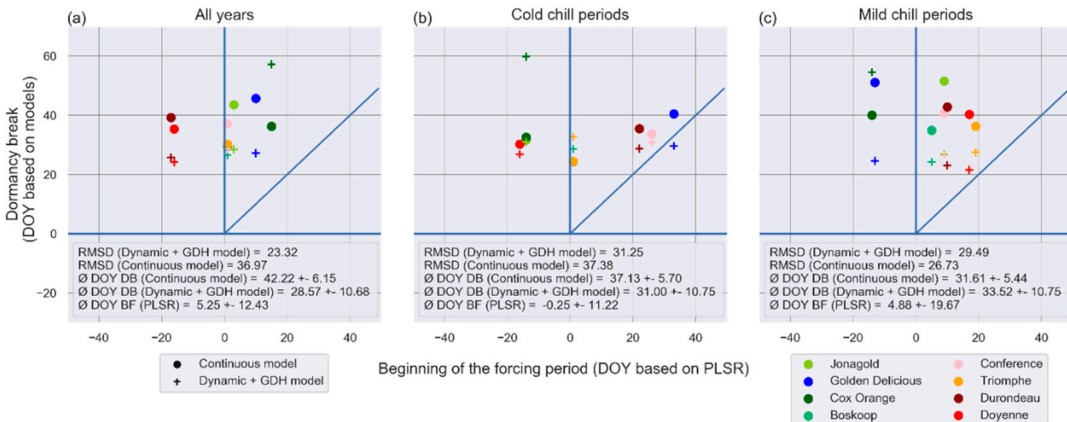

**Figure 6.** Day of the year (DOY) of the cultivar specific dormancy break (DB) simulated by the continuous model and the dynamic + growing degree hour (GDH) model, against the beginning of the forcing period (BF) estimated using the PLSR method, for all years (**a**), the 50% cold (**b**) and 50% mild years (**c**), as defined by the average temperature during October, November, and December. Symbols distinguish between used models. The diagonal lines represent the identity line, X = Y, horizontal and vertical lines highlight DOY = 0 as reference. The root mean square difference (RMSD) value corresponds to the average RMSD between the model output and the PLS results. Ø DOY gives the average specified DOY for the given years with one standard deviation.

### *3.3. Phenological Models: Sensitivity of Predictions to the Model and to the Parameter Calibration Algorithm*

Models were fitted using six different optimization algorithms, of which the best has been retained for every cultivar, based on the lowest prediction error. The resulting values are given in Tables A1 and A2 of the Annex. In the case of 'Conference', for instance, the DEMC-z performed best. Over all cultivars, the root mean square error (RMSE) of the predicted bloom date for the best fit parametrization varies between 5.36 days and 6.26 days for the continuous model and between 4.15 and 6.36 days for the dynamic + GDH model. At 99% confidence level, the RMSE for all tested parametrizations and cultivars is significantly lower for the dynamic + GDH model. For the latter, the difference between the errors of the calibration and validation subsets (1.68 days) was higher than for the continuous model (0.15 days).

How the continuous model and the dynamic + GDH model predict flowering per year for 'Conference' is exemplarily illustrated in Figure 7. The coloring allows one to identify the relative temperature during the full dormancy period (ONDJFMA), where dark dots represent cold and bright dots mild years.

Not all years and cultivars were equally well predicted, and the modelled bloom dates in cold years were generally better correlated with the observations. The ratio of the RMSE and the mean absolute error (MAE) of the mild 25% years over cold 25% years (values in brackets) indicates, in most cases, higher errors in the mild years. From a year by year comparison over all cultivars and algorithms, it emerged that the early 2000 years are largely underestimated by both models and that only few years (2004, 2005, and 1975) have consistently failed for both models (mean absolute error ≥ 7 days). Out of the 5892 modelled years, for five years, no value was returned and for two years no flowering in spring but a date in autumn was returned. In these years, for the given parameters, chilling has not been fulfilled. This shortcoming is only observed for the continuous model, mostly related to exceptionally warm October, November, and December temperatures preceding the flowering of 2007 and 1990. One such point is not shown in Figure 6c, which is the origin of the high error.

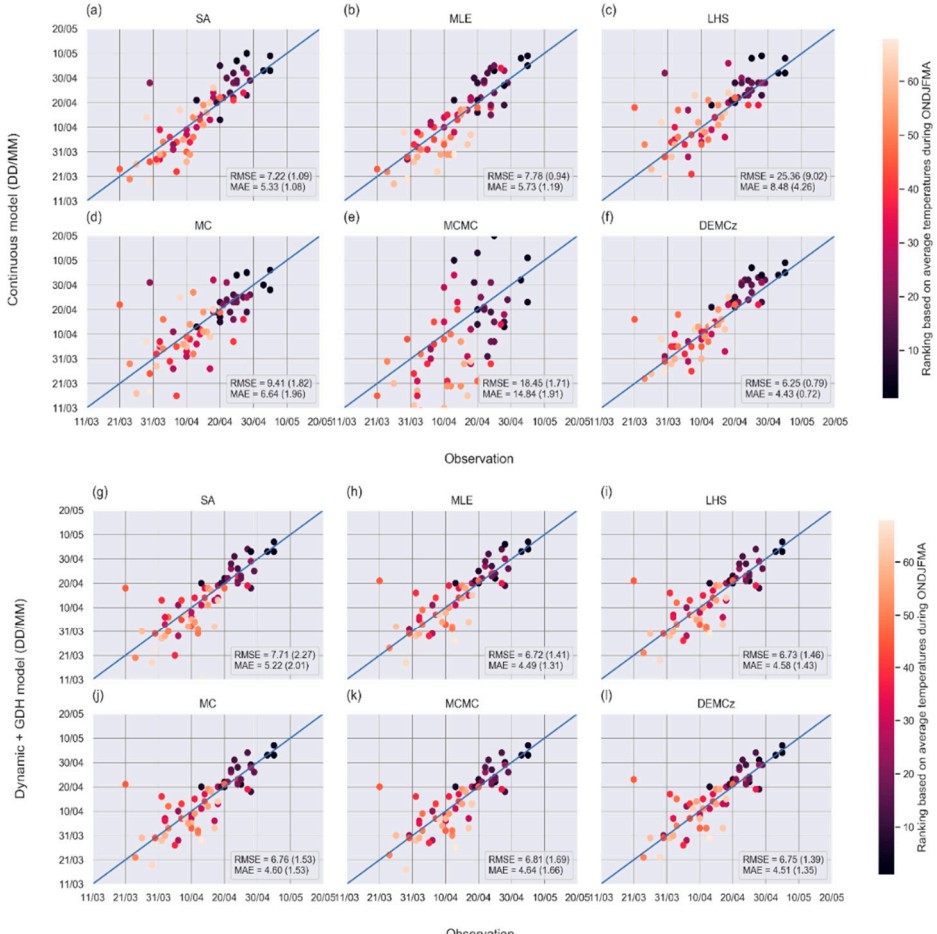

**Figure 7.** Performance of the continuous model (**a**–**f**) and the dynamic + GDH model (**g**–**l**) for 'Conference' to predict bloom date. Colors represent the position in the ranking based on average temperature during the full dormancy phase (October to April), with dark tones representing the cold years. RMSE and MAE values refer to the entire period, the values in brackets give the ratio of the respective error for the warm 25% of the years over the cold 25% of the years. For improved visibility, some points are omitted. As reference, the identity lines are shown (x = y, in blue).

In Figure 8, the ratio of performance to interquartile distance (RPIQ) is given, where higher values are desirable [41] since it implies that the RMSE is smaller compared to the interannual variability of the bloom date. In this representation, one can see that the more complex optimization algorithms (SA and MCMC) perform less well and the continuous model led to more underperforming parametrizations (strong outliers) than the dynamic + GDH model. While the differences in terms of the RMSE between apple and pear are less than half a day for both models, in terms of RPIQ, they perform better for pear than for apple. This is due to higher inter quartile range (14–17 days for pears; 12 days for apple except for Boskoop (16 days)).

Based on the RMSE and MAE (Figure 7) as well as the RPIQ (Figure 8), the dynamic + GDH model is better performing, confirming the initial hypothesis. It also has a lower spread of errors, no strictly failed predictions (i.e., return 'no value'), and higher agreement between the optimization functions than the continuous model, implying it is less sensitive to the method used for calibration and more comparable across studies.

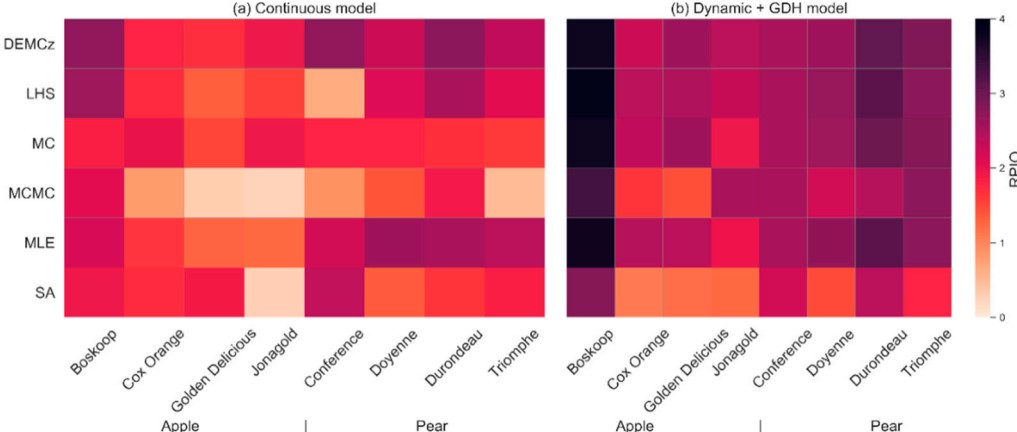

**Figure 8.** Ratio of performance to interquartile distance (RPIQ) for all cultivars and algorithms for the continuous model (**a**) and the dynamic + GDH model (**b**). Mean indices were 1.6 for apple and 1.9 for pear using the continuous model; and 2.4 for apple and 2.6 for pear using the dynamic + GDH model.

## 4. Discussion

### 4.1. Past Change Signals

Segmenting 68 years of phenological observations into contrasting segments allowed to compare flowering under relatively cold and mild dormancy periods for apple and pear. The determining phase for changed bloom timing was the forcing phase (January to April), with a significant negative relationship between temperature and the DOY of flowering, which has also been reported for other temperate regions [13,42]. Effects of a warm and a cold year compared to a typical year have been studied for mountainous forest trees [43] with similar results regarding the magnitude of the effect of warming during the winter and spring months. The present study allows for more robust conclusions due to the longer data series.

From the past observations, it appeared that the phenology of pear is more strongly affected by warmer temperatures than apple, which could be related to lower heat requirements of pear, but further physiological research is needed to explain the difference in sensitivity.

Although some authors assume linear relationships between temperature and phenology [11,44], hints for antagonistic effects could be detected at our study site, revealing non-linear correlations (Figure 3), similar to findings for other temperate regions [42,45,46]. It also appeared that the relative influence of temperatures during chilling increases with the average temperature of a location [34].

Despite pronounced warming, two of the last 10 years were among the coldest years of the whole time series. This underlines that a risk for destructive frosts remains present and warrants further research.

Phenological Response to Warming: Partial Least Square Regression

Based on the analysis of all years, Figure 4 shows a clearer average timing of the transition from chilling to forcing periods per cultivar than Figure 5, where less years were considered. This study confirmed that the number of years had a major impact on the patterns of the PLSR results, as also concluded by [10], where the detection of the transition point was not possible for observations <20 years. A study based on 15 years [34] also presented unclear and ridged curves, underlining that short time series with less natural variability compromise the functioning of PLSR. The 34 years of this study yield in most cases satisfying results, suggesting a limitation of this approach to cultivars which are grown for more than ~30 years.

In nearly all cases (34 or 68 years considered), signals in December and January varied strongly and no clear trend was observed, which suggests an overlap of chilling and forcing [10]. In a global

comparison of phenological models for apple fruit, overlapping chill-force models outperformed a strictly sequential approach [47].

### 4.2. Shifts in Phenological Phases

Delays of dormancy breaks have been reported to date for Mediterranean and Japanese settings [14,15,48,49]. For Germany, rather stable dates of dormancy fulfilment for the past years and delayed or no fulfilments for future years were reported, with dormancy break dates between DOY −20 and 20 for apple cultivars with low chilling requirements (700 Chill hours) as well as a high inter annual variability [20]. PLSR on cherry bloom observations in Klein-Altendorf (Germany) revealed phase transitions around end December and similar fluctuations in January [33].

For our study site in east of Flanders, Belgium, the chilling phase appeared to end in the last weeks of February, which could be due to a comparatively mild Atlantic climate. PLSR analyses in warmer environments reported chilling of apple to last until 19 February in China, with an annual average temperature of 11.5 °C, thus 1.4° warmer than on our study site in Belgium [34]. Chilling phases of Walnut in Davis (California) were found to last on average until 15 January. The findings of the PLSR appear reasonable, since Figure 6 illustrates that in all but one case ('Golden Delicious') the modelled dormancy break occurred later than the beginning of forcing derived by PLSR. While the two models yielded surprisingly similar results when chilling was high (cold October, November, and December, Figure 6b), they diverge when conserving all years (Figure 6a) or years with milder October, November, and December (Figure 6c). This indicates overlapping processes and that the beginning of forcing (even as dominant process) does not imply that the chilling requirement is fulfilled.

In general, the differences between cold and mild years are small. This is not surprising in light of the small changes in terms of effective chilling hours (Table 3). This can also be because a lower boundary for effective chilling is applicable, which is more often reached with rising temperatures. In other words, warmer climates can in some cases lead to more chilling if temperatures are less frequently below the threshold to contribute to chilling.

### 4.3. Evaluation of the Phenological Models

Generally, the development of regional models without field observations of dormancy break has strong limitations [50]. The choice of the model has been shown to be among the biggest factors influencing the model's outcome [9,51] and models cannot be applied across regions without further adaptation. In this study, two contrasting models were therefore calibrated to the Belgian context, where the dynamic + GDH model outperformed the continuous model, as hypothesized.

The dynamic chill model has the advantage that accumulated chilling can be partially undone by subsequent high temperatures, while the continuous model only differentiates between chill accumulation at effective temperatures and stagnation at ineffective temperatures. However, the dynamic model combined with the GDH accumulation in this setup does not allow one to account for overlap of chilling and forcing [47], nor interaction terms between respective requirements [38], which calls for improved combined models that cover the full dormancy cycle. Another of its strong points is, however, that slightly negative temperatures account for chilling, in contrast to the tested continuous model as adapted to German conditions [8]. Experiments on cherry twigs point to chilling effectiveness below 0 °C [18]. The dynamic + GDH model used hourly data while the continuous model relied on daily observations alone, but [50] found that using hourly data did not improve the modelling effort by more than 0.24 days of RMSE. Computational expenses are higher for the dynamic model than for the continuous model, but for the studied extent, this is not an important limiting factor. The processes behind the dynamic model are, however, also less transparent and flexible than the continuous model, which can be reproduced from the equations given in this manuscript alone.

For every cultivar, one best fit model setup has been selected (Tables A1 and A2). However, since none of the tested parameter estimation algorithms consistently led to the best results for all cultivars, we suggest that a set of algorithms rather than one single algorithm can be used as model ensemble

to better cover uncertainties in a similar way to climate model ensembles. According to [8,19,52–54], simulated annealing (SA) is preferred, whereas in this study SA only performed best for 'Golden Delicious' (continuous model).

In terms of RMSE, the prediction of bloom dates is of similar quality to that in the literature [20,47,55], which is commonly judged as acceptable [8]. However, when the values were compared to the variability of the dependent variable (date of bloom), using the RPIQ the performance of both models appears weak in the majority of cases (~RPIQ < 3) since this implies the natural variability of the phenomenon is less than three times larger than the RMSE.

Prediction errors in studies with long time series for individual cultivars are higher than in studies using shorter time series [11,44,56] or unspecified cultivars [8]. Together with the observation that events in mild years are mostly less well predicted than in cold years, this suggests that the models have a limited temperature interval for which they were developed and in which they performed best. This sheds doubt on their capacity to perform in future warmer regimes.

Models can be improved with dormancy break observations, but to date, only a limited number of controlled experiments on fruit and nut trees allowed for calibrating of models against dormancy phase change observations [6,18,52,57]. New research is emerging using experimental indicators such as water content or metabolites such as abscisic acids or sugars to assess the dormancy state [58–60], but the number of years of observations is still too low for long-term analyses, so alternative techniques like the PLSR are preferred. In the absence of local field experiments to determine dormancy break, phenological models can only provide indications of phenological trends.

## 5. Conclusions

In this study, 68 years of observations of bloom date of eight apple and pear cultivars have been analyzed with respect to the average temperature during the periods of full dormancy, chilling, and forcing. Advances of bloom dates with warmer dormancy periods have been found to be stronger for pear than for apple. From the observations and the PLSR, the temperatures during the forcing period appeared to have a stronger influence on bloom date than temperatures during the chilling period. However, a non-linear relationship between temperatures and bloom timing and antagonistic trends were detected in relation to increasing temperatures during late autumn months (October to December, i.e., the chilling period). For the warmest dormancy periods, which coincide largely with the more recent years, delayed fulfilment of chilling requirements has been modelled.

The PLSR method is inherently limited to long time series but was found to be promising to assess timing of phase change between chilling and forcing. The identified dates for the beginning of forcing differ from the modelled dormancy breaks. This supports the idea that chilling and forcing processes occur simultaneously.

In terms of the RMSE, MAE, and RPIQ indices over all the years, as well as regarding the spread of absolute yearly errors, the investigated dynamic + GDH model performed better than its continuous counterpart. Whereas for climate change studies it is important that the model be consistent outside of the calibration ranges, this is not apparent from the studied 68 years.

**Author Contributions:** Conceptualization, methodology, funding acquisition: A.G. and J.V.O., B.D., validation, formal analysis, writing—original draft preparation: B.D., resources, data curation: S.R.; writing—review and editing, A.G., J.V.O. and S.R., supervision: A.G. and J.V.O. All authors have read and agreed to the published version of the manuscript.

**Funding:** This research was funded by Fonds Wetenschappelijk Onderzoek Vlaanderen (Belgium), grant number 49332.

**Acknowledgments:** Research Centre for Fruit (Pcfruit vzw), 3800 Sint-Truiden (Kerkom), Belgium; especially Dr. Serge Remy and ir. Wim Verjans for the trustful provision of long time observations of phenological stages and temperatures.

**Conflicts of Interest:** The authors declare no conflict of interest.

## Appendix A

**Table A1.** Parameters from optimisations with the lowest RMSE phase, continuous model. The RMSE values are given separately for the calibration and validation sets. The mean column gives the mean of both sets.

| Cultivar | Algorithm | C | TBc | TBf | Tup | a | b | Calibration | RMSE Validation | Mean |
|---|---|---|---|---|---|---|---|---|---|---|
| Boskoop | DEMCz | 41.84 | 2.00 | 5.00 | 10.87 | 140.00 | −0.001 | 6.28 | 5.43 | 5.85 |
| Cox Orange | MC | 54.10 | 2.91 | 5.74 | 12.64 | 162.00 | −0.00584 | 5.90 | 6.15 | 6.03 |
| Golden D. | SA | 48.44 | 3.61 | 4.77 | 10.38 | 170.00 | −0.00392 | 6.31 | 6.47 | 6.39 |
| Jonagold | DEMCz | 40.00 | 4.69 | 5.00 | 9.17 | 192.20 | −0.00736 | 7.32 | 4.80 | 6.06 |
| Conference | DEMCz | 59.44 | 2.00 | 4.61 | 13.95 | 214.50 | −0.01 | 4.15 | 7.76 | 5.96 |
| Doyenne | MLE | 50.00 | 2.00 | 4.68 | 12.14 | 214.20 | −0.01 | 4.98 | 5.73 | 5.36 |
| Durondeau | DEMCz | 62.44 | 4.65 | 4.71 | 14.00 | 214.20 | −0.01 | 6.02 | 6.34 | 6.18 |
| Triomphe | MLE | 42.72 | 3.84 | 4.60 | 10.51 | 203.80 | −0.011505 | 6.55 | 6.03 | 6.29 |

**Table A2.** Parameters from optimisations with the lowest RMSE, dynamic + GDH model. The RMSE values are given separately for the calibration and validation sets. The mean column gives the mean of both sets.

| Cultivar | Algorithm | CR | HR | Calibration | RMSE Validation | Mean |
|---|---|---|---|---|---|---|
| Boskoop | LHS | 59.16 | 4430 | 4.10 | 4.20 | 4.15 |
| Cox Orange | LHS | 79.6 | 4268 | 4.21 | 5.70 | 4.95 |
| Golden Delicious | MC | 59.84 | 4980 | 4.08 | 5.12 | 4.60 |
| Jonagold | MCMC | 60.66 | 4644 | 4.85 | 4.57 | 4.71 |
| Conference | DEMCz | 60.6 | 3484 | 4.48 | 8.39 | 6.43 |
| Doyenne | MLE | 57.62 | 4184 | 3.54 | 6.35 | 4.95 |
| Durondeau | MLE | 58.66 | 3464 | 3.88 | 6.50 | 5.19 |
| Triomphe | DEMCz | 61.38 | 3632 | 4.48 | 5.98 | 5.23 |

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
