# Peer review of "Comparing Apple and Pear Phenology and Model Performance: What Seven Decades of Observations Reveal"

_agronomy, doi:10.3390/agronomy10010073_

Round 1
Reviewer 1 Report
The manuscript is very interesting and well presented. Original is the approuch of data analysis. Results support conclusions. Authors should indicate what represent the lines in fig. 7
Author Response
Response to reviewer 1
Dear reviewer,
We thank you for giving us the opportunity to further improve our manuscript. Please find the revised version of our article “Comparing Apple and Pear Phenology and Model Performance: What 7 Decades of Observations Reveal”. We greatly acknowledge your comments and suggestions. Below, we provide an extensive answer to these comments.
With kind regards,
Bianca Drepper and co-authors
Point 1 : The manuscript is very interesting and well presented. Original is the approach of data analysis. Results support conclusions. Authors should indicate what represent the lines in fig. 7
Response 1: Thank you very much for your positive feedback, it is very much appreciated. We have followed your advice to indicate the meaning of the lines in fig. 7. These lines represent the identity line (x = y). It is a good point since the lines could be confused with a trendline or line of best fits. Consequently, we also added this information to Figure 6.
“As reference the identity lines are shown (x=y, in blue).”, in ll.454 - 455 of the revised manuscript and “The diagonal lines represent the identity line, X=Y, horizontal and vertical lines highlight DOY=0 as reference.”, in ll.420 -4221.
Reviewer 2 Report
This manuscript combines two phenological model, which are parameterized to the local datasets, and a PLSR approach to analyse a long term blooming dataset of four apple and four pear cultivars grown in northeast Flanders.
The manuscript is well written and presented. There are some minor points that deserve clarification, which are marked in the attached file.
My main concern with the manuscript is the assumption that PLS determines the dormancy braking point: most of the recent literature has used this statistical technique to independently delineate chilling and forcing phases and, as a result, overlaps and gaps have been detected between both phases. The way it is used in this manuscript, PLSR detects the beginning of the forcing phase. Hence, it seems more appropriate to refer to it as a transition between phases and not as dormancy breaking point, in the case of PLSR. As a consequence, some (all?) of the discrepancies between the two phenological models and the PLSR approach may be due to the fact that they are measuring two different although related things. Authors should include this problem into their discussion, change some expressions accordingly in materials and methods and in results, and elaborate a bit on it. This includes a pint the authors have avoided: why are they using PLSR on temperature data and not on chill and heat units as the more recent papers on the subject do? Of course this is legitimate, but they should expose their reasons.
Some papers not included in the manuscript, such as other papers of Benmoussa et al on almonds and pistachio, or of Guo et all on chestnut and jujube could help on that. In this point I should mention that I have included in the manuscript a reference to a very recent paper in the same Special Issue that this manuscript is intended for, and the the authors could not be aware of when writing this manuscript: as I mention there, I believe it may help in some points as it has points of contact with this manuscript, but the authors need not feel they are forced to use it if they do not consider it appropriate.
This also means that the overlaps detected in this manuscript are not a novelty, but a confirmation that it is not confined to warmer climates.
A bit more of comments in the attached file

Author Response
Response to reviewer 2
Dear reviewer,
We thank you for giving us the opportunity to further improve our manuscript. Please find the revised version of our article “Comparing Apple and Pear Phenology and Model Performance: What 7 Decades of Observations Reveal”. We greatly acknowledge your constructive comments and suggestions. Below, we provide an extensive answer to these comments and discuss the most important changes that have been introduced to the text. All comments are listed, page by page (of the original document). The line numbers refer to the revised version (track changes mode ‘Simple Markup’ using Microsoft Word). We included the added or revised sentences in our response in bold font when relevant.
With kind regards,
Bianca Drepper and co-authors
Page 1
Point 1.1 I gues the authors mean averages flowering dates were, respectively, 9.5 and 11.5 days earlier ... or any other gramatically correct construction. Or do authors mean versus (vs.)?
I am not familiar with this use and I find the use of vs. very common, so I would advise to change it throughout, but I am not an expert in the matter.
Response 1.1: The usage of ‘resp.’ is common in mathematic texts but in most other cases it seems indeed to be not in common use. The usage of ‘resp.’ has thus been avoided and reformulated throughout the document.
13-14: Median flowering dates of apple were on average 9.5 days earlier following warm dormancy periods, and 11.5 days for pear, but… 150-152: Therefore, contrasting periods were selected based on rankings of mean temperatures during the full dormancy periods (October to April incl.), chilling periods (October, November and December) and forcing periods (January to April incl.) 198: the 34 cold and 34 mild years 267-268: The parameter distribution was assumed to be uniform for the required parameters, resulting in spaces as described in Table 1 and Table 2. 276: Finally, 500 or 200 iterations were called 299- 301: For apple vs. pear, resulting advances of the median flowering date were 9.5 vs. 11.25 days (12.5 days for ‘Conférence’, Figure 3 (a)), and 10.3 vs. 13.38 days (14.5 days for ‘Conférence’, Figure 3 (c)). 301-303: In contrast, with 0.5 days delay and 1.6 days advance (median), no significant change was detected when temperatures during October, November and December determined the segmentation, Figure 3 (b). 321-324: Blooming occurred significantly later during the medium years (at 99% conf. level) than during the cold 25% years: on average 13.0 days (apple) and 13.6 days (pear). The difference between the mild 25% years and the medium 50% years was 2.75 days for apple and 1.0 days for pear; none of these values were statistically significant. 324 – 325: However, the difference in mean temperature between the first and second (1.36°C); and, second and third segment is nearly identical ( 1.37°C). 327 – 330: The decomposition of the analysis into temperatures during chilling and forcing also revealed disparate patterns. The relationship of rising temperatures during January to April and bloom date was consistently and strongly negative (17.5 days advance from the cold to the medium and 20.75 days from the medium to the warm segment). 336 -338: The share of effective hours was reduced by 0.95 – 2.58 % if the years were split in two halves, with the more pronounced reduction in the most efficient interval (3-9°C); and by 2.39 - 4.31% when the cold and mild 25% of the years were considered (Table 3). 469-471: Mean indices were 1.6 for apple and 1.9 for pear using the continuous model; and 2.4 for apple and 2.6 for pear using the dynamic + GDH model.
Point 1.2 a semicolon is lacking here?
Response 1.2: Yes, indeed. A semicolon has been added in l.24.
Point 1.3 This nomenclature may be a source of confusion as the term 'sequential model' (here related to a coupling between chill and forcing phase with an exponential relationship between chill and forcing requirements), has often been used in the literature to refer more generally to a combined model in which a chill phase is immediately followed by a forcing phase without any overlap nor discontinuity, and irrespective of the way each phase is determined (be it the dynamic model and chill portions, Utah model and chill hours or any other chill unit). In the same context, the dynamic model refers only to the chill phase and is followed by a forcing phase usually related to GDH, as here. It would be preferable using a safer naming, such as coupled or linked model... vs Dynamic+GDH..., nonsequential Dynamic+GDH... NSDG... In fact, if I understood it correctly, the 'dynamic' model here is used sequentially also
Response 1.3: Thank you for this helpful remark. We agree on changing the naming considering that indeed both models are characterized by a sequential set up in this study. We originally chose the name ‘sequential model’ to be in line with the referenced literature (Chmielewski et al., 2011) and used the shortened name ‘Dynamic model’ for improved readability.
It proved to be not intuitive to replace ‘sequential model’ by a synonymous label like ‘linked’ or ‘coupled model’, since one may argue that both models are linked or coupled, but in different ways. To mark the distinction between both models, the adjective ‘continuous’ has been chosen to characterise the processes behind the model, as opposed to dynamic. In the model previously called ‘sequential model’ chilling accumulates or stagnates, but is never reversed.
Hence, throughout the manuscript, ‘Sequential model’ has been consistently replaced by ‘Continuous model’ and ‘Dynamic model’ by ‘Dynamic + GDH model’, when the combined set-up is being referred to. The introduction to the models is adjusted as stated below but further individual replacements are not listed in this document since they do not alter the interpretation of the corresponding phrases.
84-101: Recent experimental studies on cherry trees in Germany [18] suggest that faster accumulation of forcing can substitute up to 50% of lower accumulated chilling, but a minimum amount of chilling remains to be fulfilled before forcing can be effective. This motivates the choice of sequential modelling approaches, that explicitly accounts for a cultivar specific relationship between chilling and forcing requirements. One such a model set-up emerged as the most adequate for climate change impact studies from an extensive evaluation of mechanistic models for the beginning of apple blossom in Germany [8] and has been applied in several studies [11,19,20]. Over time, first chilling and consequently forcing accumulate or stagnate, but never reverse. This continuous, undynamic, approach has the advantage of fast and transparent computations, that allow for easy adaptation to different fruits and local conditions, but results in less reliable estimations of chilling requirements [21]. Hence, the continuous model only leads to reasonable results within a narrow range of temperatures, where chilling is fulfilled before an accumulation of Growing Degree Hours is possible.
The chilling and subsequent forcing expressed in this continuous model are thought to hold for the past in Belgium, but seem inadequate in a warming world. Reviews of chill and phenology models [5,9,22] suggest a more complex dynamic process [6,23], that accounts more realistically for chilling and forcing processes. We hypothesised that dynamic phenological modelling captures the chilling and forcing requirements in a better manner, and can be used to simulate phenological development of apples and pears in a warming climate.
Point 1.4 There is some contradiction here: weather effects lead to lowe production and hence hihgher prices, and Russia Embargo to the contrary: lower global demand, excess relative production and price drop.
Response 1.4: We agree with your reasoning in terms of demand and supply. The Russian embargo is regarded as the primary problem of the sector by the farmers themselves (Avermaete et al., 2018), and has thus been mentioned at this place of the manuscript. It was however only after several weather related production shocks in the previous years that the sector has been identified as a sector in crisis, hence the mention of the combined factor. For clarity a reformulation has been attempted as below:
41-43: The consequences of extreme weather in addition to the Russian embargo on fruit imports since 2014 led the entire Belgian fruit sector into an officially recognized crisis[.]
Page 2
Point 2.1 I agree with the general meaning of this sentence. However, I feel authors should be more precise: the fact that a better knowledge of climate change impacts on apple and pear crops in Flanders is needed does not imply that the effects of warming on phenology is not already well known. In fact, the authors apply models previously developed that might work well enough...
Response 2.1: This is a good point. Since the effects of warming on phenology are location dependent we specified that better knowledge of this effect is needed for the conditions in Flanders as follows:
46-49: Therefore a better understanding is needed of the impacts of warming on phenology in the specific case of apple and pear cultivation in Flanders. By means of adapted phenological models, variations in flowering and the related frost-sensitive stage can be estimated across temporal and spatial scales, which can support decision-making regarding (new) orchard locations and cultivar selection.
Point 2.2 Please be consistent: in most instances, you have used "parametrisation"
Response 2.2: The correction “parametrisation” has been applied.
Point 2.3 [11] (Funes et al 2016) does not use PLSR but simple regressions in their approach!
Also, I have been hesitant to make this comment to avoid any hint of ethical misbehaviour, so I will say it plain: from the same authors of 11, a new paper has just been published in the same Special Issue of Agronomy. Of course, the authors of this manuscript were not aware of it as it was publised so recently that ther manuscript was most probably sent to Agronomy before its publication. However, I dare to mention it because it mentions a possible limitation of PLSR approach, in the sense of it beeing a good tool to deal with limited observantions in front of large number of predicting variables, but maybe not good enough for the case of phenology, where chosing the relevant variables in not enough, as it implies choosing also other variables (contiguous days) that are not originally chosen by PLS. So, the authors of this manuscript might feel that they could include this reference here or in the discussion. But if they decide not to do so, it would not be at all detrimental to this manuscript. The reference would be:
Díez-Palet, I.; Funes, I.; Savé, R.; Biel, C.; de Herralde, F.; Miarnau, X.; Vargas, F.; Àvila, G.; Carbó, J.; Aranda, X. Blooming under Mediterranean Climate: Estimating Cultivar-Specific Chill and Heat Requirements of Almond and Apple Trees Using a Statistical Approach. Agronomy 2019, 9, 760.
Response 2.3: Thank you for bringing this interesting publication to our attention. We have opted not to replace a present reference with the suggested reference in our manuscript. The reference (Funes et al., 2016) has however been removed from this location (l.66).
Point 2.4 : Wouldn't this minimum be the true chill requirement? Wouldn't htis substitution be only apparent, and the true relationship based the raionale exposed in the previous paragraph? If a blooming date is modeled, changes in the time needed to complete both the chill phase and forcing phase are to be expected if T rises (longer and shorter, respectively), compensating to each other, which might look as substitution of chill and forcing requirements, depending on how those requirements are determined.
This belongs to the discussion, probably, but I wrote this here as I was reading: excuse me if I did not delte or move it to the appropriate section
Response 2.4: We understand your point and added a clarification to the manuscript as this sentence was meant to state that despite a substantial substitution potential, a certain chilling requirement must still be met.
Ll 83-84 : Recent experimental studies on cherry trees in Germany [18] suggest that faster accumulation of forcing can substitute up to 50% of lower accumulated chilling, but a minimum amount of chilling remains to be fulfilled before forcing can be effective.
Point 2.5 hypotesyse
Response 2.5: Unfortunately it is not clear to us what this comment is meant to tell us. The spelling of ’We hypothesised’ in l. 99 of the manuscript is correct British English. Please clarify if need be.
Page 3
Point 3.1 Please describe it better (e.g. as in the acknowledgments)
Response 3.1: Full details have been given:
120-123: The timing of the phenological stages according to the definitions of the Biologische Bundesanstalt, Bundessortenamt and CHemical industry (BBCH) [24] has been recorded since 1950 for main apple and pear cultivars by the Research Centre for Fruit (Pcfruit vzw), Sint-Truiden (Kerkom), Belgium.
Page 4
Point 4.1 Please rephrase: the differences were in temperature before and after 1988, in the impacts of temperature, in the general impacts of climate chage and a temperature difference between
both periods is responsible of these differences, ...?
Response 4.1: Thank you for this remark. The sentence has been rephrased for clarity:
141 -144 A study of climate impact on growth and productivity of arable crops in Belgium showed significant differences in temperature before and after a change point in 1988 [28], which allowed to compare weather impacts on the growing season and on sensitive phenological stages for two contrasting climatic periods.
Point 4.2 I must be missing something or the authors are not clear enough in this point, but I do not see the relationship between this part of the paragraph (nor the end of the previous paragraph)
with the second part, which is a common procedure that need not much previous justification
Response 4.2: Thank you for your remark. The paragraph has been rephrased below for more clarity.
ll.144 – 152: Change point analysis was also applied directly to phenology data for West Germany, Switzerland and the Balkans [29–31] and suggested a break point around 1988.
However, for different temperate locations, 2-segmented pairwise constant models with such a breakpoint and used to correlate temperature and phenological stages did not perform consistently better than linear models according to the modified Bayesian information criterion [13]. This suggests that complex processes such as phenological development need further analysis to disentangle effects of warming on chilling and forcing. Therefore, contrasting periods were selected based on rankings of mean temperatures during the full dormancy periods (October to April incl.), chilling periods (October, November and December) and forcing periods (January to April incl.).
Point 4.3 Isn't a reference lacking here?
Response 4.3: Yes indeed, this reference was missing. I refer to the temperature indications by (Erez et al., 1989). The reference has been added to the manuscript (l.162).
Point 4.4 Why not on the chill and heat accumulated, as in [33]? I beleive authors should justify their choice to use PLSR on temperature, given this is not how chillR seems to be mostly used in the
last years: it seemed a step forward not to use temperaute values but something more related to its effect on the plant. Although this introduces the need to choose a function than
corresponds to this effect (i.e. tranforming it in chill portions, chill hours, chill units... and same for GDH), abandoning it deserves a justification
Response 4.4: It has indeed become more common to use the PLS on daily chill instead of temperature as independent variables. However, in this study it would not allow to compare the output of both models in the same way, if the independent variables of the PLSR are the result of one of the two compared models. Results based on daily chilling are also much noisier, than results based on temperature, as for instance in (Guo et al., 2019, figure 3). Also, working with temperature reduced the number of panels to consider in this extensive comparison study. Finally, from the suggested study (Díez-Palet et al., 2019) limitations of the PLSR ran on daily chilling and forcing became apparent, especially for determining the end of the chilling phase. A justification has been added:
177-179: Daily temperature, rather than modelled daily chilling and forcing accumulation, was chosen as independent variable to enable the comparison of the model-independent output with both selected phenological models.
Point 4.5 Does PLSR confirm or contradict this assumption?
Response 4.5: Thank you for your remark. We have run the PLSR (using all years) on longer annual timeseries (starting 1.August) and found that the derived dates were the same ( or +- 1-2 days in 5 out of 8 cases; in 6 out of 8 cases they coincided if the PLSR was considering days after 1st-September or 1st -November). The other cases did not differ much visually but other days were detected based on the defined selection criteria that were refined for the output starting in October. We concluded that the sensitivity to the starting day is low.
Because we consider contribution to chilling in August and early September an artefact of the model to be avoided, we chose 1st October as it comes closer to the time that endodormancy starts without losing potentially crucial days. This way, readability is increased as well.
173 – 176: In this application, the observed flowering date per year [Julian Day of Year (DOY)] was related to the antecedent temperatures [°C] on a daily basis. The beginning of the dormancy period has been approximated as 1st –October, after finding the sign change analysis not to be very sensitive to the starting date. This way, model artefacts, i.e. chilling contribution in August, were reduced.
Page 5
Point 5.1 Irrespective of later appearance of positive coefficients with VIP>0.8 as in all pear cultivars and also in Junagold and Cox Orange? This is a bit counterintuitive to me, especially in the case
of Durondeau and somewhat in Doyenné. According to several of the references cited here (mostly from Luedeling and colleagues), this would corrspond to the beginning of the forcing
phase, not exactly to dormancy break. in fact, if an overlap is accepted, as seems to be exposed later, there is not exactly a transition form the chill to the forcing phase, as they overlap.
Maybe it could be more accurately referred as the beginning of the transition from chilling to forcing phase, or more asseptically, the beginning of the forcing phase. In this case, the end of
the forcing phase should also be defined here (as the end of "green" periods).
Response 5.1:
We agree that an interpretation of the transition point using the PLSR as in this manuscript is more intuitive as the beginning of forcing. This differentiation allows to explain the discrepancy exposed in Figure 7.
Accordingly we have adjusted the naming, interpretation and conclusions throughout the document. Numbers changed since also only the best fit models were considered.
16-18: After warm chilling periods, an average delay of 5.0 and 10.6 days in the occurrence date of dormancy break was predicted by the phenological models, while the PLSR reveals mixed signals regarding the beginning of flowering. 102 -108: The major objectives of this paper are 1) to analyse phenological responses to increasing temperatures during chilling and forcing periods, with the hypothesis that bloom dates are not proportionally advancing; 2) to detect shifts in the beginning of the forcing phase using Partial Least Square Regression and compare these shifts with modelled dormancy break dates and finally, 3) to quantify chilling and forcing requirements for a range of cultivars of apples and pears at a study site in Belgium, while comparing the selected continuous and dynamic phenological models and their sensitivity to parameter optimisation. 198 -202: Using the tabular output of the PLS function, the date of the beginning of forcing is inferred as the first occurrence of a sign change from a positive to a negative coefficient, followed by (i) at least three days out of 4, 7 or 17 days (iterative process) with the occurrence of a VIP value > 0.8 and a negative coefficient, and (ii) a median negative coefficient for the following 10 days. Only dates after 1st of December were considered. 348 -350: The persistent change from a positive to a negative coefficient is interpreted as a change in dormancy phases, from endodormancy to ecodormancy, further referred to as beginning of the forcing period. 364-365 (caption figure 4): Date labels indicate the beginning of the forcing period. 394-395 (caption figure 5): Date labels indicate the beginning of the forcing period.
The section 3.2. Shifts of phenological phases has been changed entirely:
397 – 423: From the PLSR results, the beginning of the forcing period can be identified from the sign change of the daily coefficients of the predictors. The earliest occurrence of significant negative coefficients has been shifted towards later dates by at least 15 days in all three rows in Figure 5. Using the average temperatures during October, November- and December as a criterion to separate cold and mild years, the days of dormancy break simulated by the continuous and the dynamic + GDH models were plotted against the beginning of the forcing periods identified by PLSR in Figure 6. The selected models are shown in Table A1 and Table A2.
Different patterns emerge from both models: under mild temperatures (Figure 6 (c) compared to Figure 6 (b) an average 10.6 days later dormancy break was modelled by the continuous model and an average of 5.0 days by the dynamic + GDH model In all cases but ‘Cox Orange’, ‘Triomphe’ and ‘Boskoop’, the dates simulated by the continuous model were found to be later. For both models, the modelled dates were consistently earlier (23.3 to 31.25 days) than the dates corresponding to the PLSR sign change, indicating the beginning of forcing. Deviation from the PLSR results were larger for the continuous model than for the dynamic + GDH model. The latter also has a larger standard deviation resulting from one outlier (‘Cox Orange’) but otherwise a smaller spread. Both models predicted as expected cases where the dormancy break approximated the beginning of forcing, especially for pear but also ‘Golden Delicious’ after cold chilling periods but the dynamic + GDH model does so more strongly after milder chill periods Figure 6 (c).
Figure 6. Day of the year (DOY) of the cultivar specific dormancy break (DB) simulated by the continuous model (CM) and the dynamic + GDH model (DM), against the beginning of the forcing period (BF) estimated using the PLSR method, for all years (a), the 50% cold (b) and 50% mild years (c), as defined by the average temperature during October, November and December. Symbols distinguish between used models. The diagonal lines represent the identity line, X=Y, horizontal and vertical lines highlight DOY=0 as reference. The RMSE value corresponds to the average RMSE between the model output and the PLS results. Ø DOY gives the average specified DOY for the given years with one standard deviation.
505: 4.2. Shifts in phenological phases
[…]
512-522: For our study site in the east of Flanders, Belgium, the chilling phase appeared to end in the last weeks of February which could be due to a comparatively mild Atlantic climate. PLSR analysis in warmer environments reported chilling of apples to last until 19th of February in China, with an annual average temperature of 11.5°C, thus 1.4° warmer than on our site in Belgium (Guo et al., 2019). Chilling phases of Walnut in Davis (California) were found to last on average until 15th of January. The findings of the PLSR appear reasonable, since Figure 6 illustrates that in all but one case (‘Golden Delicious’) the modelled dormancy break occurred later than the beginning of forcing derived by PLSR. While the two models yielded surprisingly similar results when chilling was high (cold October, November and December, Figure 6 (b)), they diverged when including all years (a) or years with mild October, November and December (c). This indicates overlapping processes such that the beginning of forcing as the dominant process does not imply that the chilling requirement is necessarily fulfilled.
[…]
581 – 584: The PLSR method is inherently limited to long time series but was found to be promising to assess timing of phase change between chilling and forcing. The fact that the identified dates for the beginning of forcing differ from the modelled dormancy breaks, support that chilling and forcing processes occur simultaneously.
Point 5. 2 According to comment in the beginning of the manuscript, it might be better to call it coupled model?
Response 5.2: As stated above, all references to both models have been adjusted based on your suggestions into ‘continuous model’ and ‘dynamic + GDH-model’.
Point 5.3 Equations 1 to 3
Response 5.3: The correction has been done in l.213.
Page 6
Point 6.1 That's one of the reasons to, perhaps, better keep this name for the chilling phase of the model and use a different name for the complete model, as suggested before.
Response 6.1: As stated above, all references to both models have been adjusted based on your suggestions into ‘continuous model’ and ‘dynamic + GDH-model’.
Point 6.2 See my comment at the beginning of the manuscript
Response 6.2: As stated above, all references to both models have been adjusted based on your suggestion into ‘continuous model’ and ‘dynamic + GDH-model’.
Point 6.3 As it is described here, it seems that authors used this model sequentially, i.e. all GDH produced after achieving CR ar counted. No room for overlap or gaps is given. (again, it would make
it confusing to call the other model "sequantial" as this model is also sequential). However, the use of PLSR would allow to use this model nonsequentially, as has been used in several
works of Luedeling and others cited in this manuscript
Response 6.3: As stated above, all references to both models have been adjusted based on your suggestions into ‘continuous model’ and ‘dynamic + GDH-model’. Because the first intention of our research was to calibrate a model to assess impacts of warmer temperatures on bloom period we considered the entire dormancy period in the modelling approach. We intend to apply the model for future climate scenarios, knowing that PLSR is not feasible due to the lack of observations.
Page 7
Point 7.1 this braket shouldn't be here, right?
Response 7.1: The bracket has been removed.
Point 7.2 [resp.]
Response 7.2: The usage of resp. has been corrected throughout the document as mentioned in response 1.1.
Point 7.3 It took me a while to get that this is the range between the first and third quartile: couldn't this be indtroduced her for clarity?
Response 7.3: The information has been added for clarity.
ll.277 - 283: […] and the Ratio of Performance to InterQuartile distance (RPIQ)(Bellon-Maurel et al., 2010), which is defined as the ratio of the range between the first and the third quartile of the observations to the RMSE of the prediction, as in equation 13. Thereby no assumptions are made about the distribution of the observed values (in contrast to the commonly used Ratio of Performance to Deviation (RPD).
RPIQ = (Q3-Q1)/RMSE (13),
where Q1 is the first and Q3the third quartile and RMSE is the Root Means Square Error.
Point 7.4 cold vs mil and cold vs intemediate vs mild? Could it be specified here for clarity?
Response 7.4: Additional information has been added. We added ‘medium’ to the legend of Figure 2.
ll.283-287: All distinct temporal segments within a panel (cold, medium and mild in Figure 2) are significantly different from each other at the 99% confidence level
Point 7.5 Why is 99% confidence level chosen instead of the more common 95%? Just curious
Response 7.5: We gave the significance level of 99% when we saw that at 95% level gave very clear results. The 99% level was meant to emphasise the clearness of the differences found. increase the accuracy of the prediction.
Point 7.6 Really? Of course it does not look negligible, but is a very smal portion (3-4 out of about 35 years?). Maybe a milder adjective would be better?
Response 7.6: Based on your recommendation the wording has been changed.
291-294 : Considering temperatures during the forcing period (Figure 2 (f)) revealed that cold springs still occurred and illustrated the important interannual variability despite the general warming of the air temperature, suggesting that a risk for late spring frost remains.
Point 7.7 [resp.]
Response 7.7: The usage of resp. has been corrected throughout the document as mentioned in response 1.1.
Point 7.8 This looks like the forcing period is the relevant one...
Response: Yes, this is true and it was mentioned in the discussion, l.475-476. This observation was also added to the conclusion
574- 580: Advances of bloom dates with warmer dormancy periods have been found to be stronger for pear than for apple. From the observations and the PLSR, the temperatures during the forcing period appeared to influence the bloom date stronger than temperatures during the chilling period. However, a non-linear relationship between temperatures and bloom timing and antagonistic trends were detected in relation to increasing temperatures during late autumn months (October to December, i.e. the chilling period). For the warmest dormancy periods, which coincide largely with the more recent years, delayed fulfilment of chilling requirements have been modelled.
Point 7.9 -11 [resp.]
Response: The usage of resp. has been corrected throughout the document as mentioned in response 1.1.
Point 7.12 Which months? OND?
Response: Yes. It has been added for clarity in l. 304.
Page 8
Point 8.1 Air temperature 2 m above soil level?
Response 8.1: Yes, exactly. The spelling has been adjusted to what seems to be most common.
311-313: Average temperature refers to daily average air temperature measured at 2m height during the indicated months at the study site.
Page 9
Point 9.1 blooming? please, clarify.
Better use a stop here, as the rest of the paragraph is not anymore about the relationship between bloom date and temperature, but about delay in blooming date
Response 9.1: The sentences have been split for clarity and ‘blooming’ has been inserted.
314-317: When looking at the median days of flowering, a nonlinear relationship is suggested between the bloom date and the temperature during the full dormancy period for the cold 25% of all bloom dates, Figure 3 (d). Blooming occurred significantly later (at 99% conf. level) than during the medium and mild years: […]
Point 9.2 This cannot be seen in figure 3 but comparing figures 2 and 3. I guess the authors did look for a linear or nonlinera relationship beteen both variables, so please mention how was is
analysed.
Response 9.2: Please refer to the response 9.1.
Point 9.3 wouldn't it be cleare to write " 13.0 days for apples and 13.6 for pears"?
Response 9.3: The formulation around the average values has been adapted according to the suggestion.
316 – 319: Blooming occurred significantly later during the medium years (at 99% conf. level) than during the cold 25% years: on average 13.0 days (apple) and 13.6 days (pear). The difference between the mild 25% years and the medium 50% years was 2.75 days for apple and 1.0 days for pear; none of these values were statistically significant.
Point 9.4 Ok, this is the rationale for the statement at the beginning of the paragraph: please, try to write it the other way around for clarity: difference in temperature between 1-2 and 2-3 is similar
but difference in bloomong dates is not, hence, a nonlinear relationship appears
Response 9.4: Indeed, we started our analysis from interpreting the full dormancy period. We have adapted the first sentence of the paragraph to reflect this starting point as indicated in our answer to point 9.1.
Point 9.5 This is something curious: when the 34 coldest years for OND are chosen, 49.69% of hours are chill effective, but when 25% coldest is chosen, this drops to 42.5%... I guess in these years
a lot of ours are below 3 celsius and are not chill efficient?
Response 9.5: Yes, exactly, this is what comes out of our research. It is mentioned in the discussion part 4.2, ll. 523 - 527. This effect is only really visible if the interval is sufficiently narrow [3,9]. Since the used models consider the mentioned larger intervals where this effect seems to be less pronounced we do not elaborate on this further. Please also refer to response 15.4.
Point 9.6 Yes, but what if the change is not persistent? The way it has been defined in mat and methods (lines 189-190) does allow for nonpersistent changes. See my comment there. [point 5.1]
Response 9.6: Please refer to the response 5.1
Page 10
Point 10.1 I wouldn't say so. To get to the conclusion the authors derive, I would point out that (a) and (c) are as similar as (d) and (f), in contrast with (b) and (e), which are similar between them
Response 10.1: The paragraph has been adjusted as below.
374-378: Figure 5 (a) and (c) resemble each other more than (b), just as in Figure 5 (d) and (f) resemble each other more than (e), implying that the warming trend during January to April dominated the ranking, similarly to Figure 3. When years were segmented based on chilling periods (Figure 5 (e) vs (b)), the coefficients are more homogenous in the mild segment, where the characteristic transition from positive to negative coefficients was more pronounced.
Point 10.2 mild?
Response 10.2: Yes, exactly. The word ‘mild’ has been inserted for clarity:
378 - 379: In these mild years, mid-December days have delaying properties (Figure 5 (e)) [.]
Page 11
Point 11.1 See previous comments about this. [probably point 5.1] Also, it seems that authors give more relevance to coefficient sign than to it VIP value, which is arguable: VIP in PLSR context can be seen as a
significance level. As much as no information value is given to nonsignificant differences in and ANOVA or to correlation nonsignificant coefficients in an ordinary regression approach, not
much value should be given to actual sign of days with low VIP, in my opinion: these may result being positive or negative at random, which would explain the alternating pattern in long
subsets of days with low VIP values
Response 11.1: The way we derived the DOY in this manuscript is considering the VIP value as criterion, acknowledging its importance for interpretation. However, as also described in (Díez-Palet et al., 2019), it is not straightforward to define a criterion that yields trustworthy results throughout the analysed cultivars and periods. We developed an approach that would search for a date with i) DOY > -25 and that ii) follows immediately after a day with a positive coefficient and which is followed by at least 3/x days that fulfil the criteria iii), having a VIP score >0.8, iv) a negative coefficient; and v) the following 10 days should have a median coefficient <0. If no point was found the search window x is increased from 4 to 7 and 17 days, to ensure a date was found. This set of criteria was established to find the date that would be visually expected but indeed there are limitations to this method.
ll.198-202: Using the tabular output of the PLS function, the date of the beginning of forcing is inferred as the first occurrence of a sign change from a positive to a negative coefficient, followed by (i) at least three days out of 4, 7 or 17 days (iterative process) with the occurrence of a VIP value > 0.8 and a negative coefficient, and (ii) a median negative coefficient for the following 10 days. Only dates after 1st of December were considered.
Point 11. 2 in this case it would not be dormancy brake but beginning of the forcing phase which, given an overlap, would be different for braking dormancy day in the models used, as they are both
sequential in the sense that both phases are coupled in a way or another
Response 11.2: Please refer to response 5.1.
Point 11.3 I guess the authors refer to the mean DOY (34 vs. 43, 36vs. 37). This might be stated more clearly.
Response 11.3: The phrase has been reformulated when the section has been rewritten, please also refer to response 5.1
ll.404- 406: Different patterns emerge from both models (regarding the mean DOY of dormancy break): under mild temperatures (Figure 6 (c)), compared to Figure 6 (b), an average 10.6 days later dormancy break was modelled by the continuous model and an average of 5.0 days by the dynamic + GDH model.
Page 12
Point 12.1 the
Point 12.2 In fact it is 0.15days if I am no wrong. Tjhis difference rang a bell on my head: the fact is that the dynamic model has consistentely a better fit (lower RMSE) in the calibration than in the
validation (except for Jonagold) , as should be expected (?), but the sequential has 3 cases out of 8 (37.5%) in which teh model fits better to the validation dataset (lower RMSE), which
looks odd. If this is taken into account and the absolute difference of RMSE is not used, but the squared root of the mean of squared of differences, this come to 2.1 days (dynamic) vs. 1.6
days (sequential), which is not so different
Response 12.2: The fact that in a few cases the validation error is lower than the calibration error is due to few individual years, failing to complete chilling with the selected parameters, occurring also during the calibration phase, with high errors that increase the mean. It is an indicator for a weak optimisation process in these particular cases. The indicated difference in the RMSE values referred to in the text as compared to the table have been traced back to the usage of a different grouping function that selected based on minimum values of the calibration error rather than the averaged error. The difference is indeed 0.15 in the case of the continuous model and 0.03 in case of the Dynamic + GDH model. The values in the text have been replaced by values corresponding to the given table.
It is not clear to us, what is meant with the suggested calculation, please clarify if applicable.
Point 12.3 Why give here all fitting algorithms just after having chosen and shown the best one for each cultivar? Way not show just DEMCz, which is the best for both models in the case of
Conférence?
Response 12.3: An updated version of Figure 6 has been limited to the best fit models per cultivar only, as specified in Table A1 and Table A2.
Point 12.4 True, but ther is something strange in pannels (a) through (c) in the values in brackets: in (a) and (b), darker points seem to be better fitted (closer to the line) than brighter points, however,
the values in brackets are <1; in (c) it is the opposite, with darker points being similarly spread as brighter ones and, ohewere, the values in brackets being quit >1. Could authors please
check that? Maybe it is becasue iof the points not shown for clarity?
Response 12.4: We recalculated the values and adjusted the plot size slightly. Mind that the order of the plots has changed. The questioned results for using algorithms MLE, MCMC (which used to be (c)) are explained when the full spread is shown as in Figure R1. The outliers in Figure R1 (b) and (e) are not fully shown in Figure 7 in the manuscript to increase detail in the other panels, while preserving the scale.
Figure R1 Extract from Figure 7 with a larger extents. Axis are given the DOY of flowering of Conference for the continuous model..
For DEMCz, on the contrary, a closer look reveals the warmest 25% years lay closest to the identity line.
One other point is shown in neither Figure R1 nor the revised Figure 7; the result for year 2007 from the LHS algorithm, with a modelled flower date of DOY = 299; indicating that for this parameter selection the chilling requirement was not met until the following autumn. The October, November and December period preceding the 2007 flowering was indeed the warmest on the ranking (see also Figure 3e). Also ‘MC’ did not yield any flowering date. The best fit model (DEMCz) however was only one day later than the observed value for this year (104), which underlines the sensitivity to the optimisation process as well as the strength of the selected algorithm.
Additional information has been added in the manuscript as follows:
442 – 447: Out of the 5892 modelled years, for five years no value was returned and for two years no flowering in spring but a date in autumn was returned. In these years, for the given parameters, chilling has not been fulfilled. This shortcoming is only observed for the continuous model in years, mostly related to exceptionally warm October, November and December temperatures preceding the flowering of 2007 and 1990. One such point is not shown in Figure 6 (c), which is the origin of the high error.
Point 12.5 Please be consistent: in most instances you hve used "parametrization
Response 12.5: The correction ‘parametrisation’ has been done in l.458.
Page 13
Point 13.1 Please revise reference
Response 13.1: The reference has been revised in l.463.
Page 14
Point 14.1 This or the present, not both
AND
Point 14.2 "longer dataset"?
Response 14.1 and 14.2: The formulations have been changed according to the suggestions.
480-481: The present study allows for more robust conclusions due to the longer data series.
Point 14.3 in
Response 14.3: The formulation original formulation ‘at our study site’ is preferred over the suggested formulation ‘in our study site’ and has not been changed (l.486).
Point 14.4 see my previous comment on that: true, this risk has not disappeared, but it is quite much lower than in past decades, to the point that growers might consider if investing in protective
structures against late frost is necessary
Response 14.4: Thank you for your remark. This sentence expresses that a risk prevails, since we did not do a thorough analysis of the evolution of the risk we chose to not make a statement on the evolution of this risk other than that it remains. For clarity this phrase has been reformulated as follows:
489– 490 Despite pronounced warming, two of the last ten years were among the coldest years of the whole time series. This underlines that a risk for destructive frosts remains present and warrants further research.
Point 14.5 Sorry, I do not get the point. Could you please rephrase, clarify...? Do you mean that the transitions are clearer in figure 4 than in figure 5?
Response 14.5: Yes, this is what we meant. For clarity, the formulation has been adapted as follows:
493-494: Based on the analysis of all years, Figure 4 shows a clearer average timing of the transition from chilling to forcing periods per cultivar, than Figure 5, where less years were considered.
Point 14.6 Mmmm, i would just say the contrary: if interannual variability is high, PLSR would be easily stablish the relevance of those days for blooming date. However, if these days have very similar
temperatures across all years, PLSR would detect no signal here and hence it would give a low VIP value.
Moreover, in a long term average it is possible that there are contradictory signals that cancel to each other, which would result in no signal at all, not in a mixture of signals.
Anyway, the fact that this behavior can be seen in the literature with quite shorter datasets, seems to point to an overlap.
Besides, would it be contradictory with the statement of the previous paragraph that PLSR would work well only with datasets including at least 30 years? I understand that, in fact, the time
span of the datasets used here is not 30 but 68 years, but maybe the authors should state this non-contradiction more clearly.
Response 14.6: This is a good point. The second suggested explanation is indeed not well justified and we chose to remove it, attributing the mixed signals to the overlapping effects.
501-502: In nearly all cases (34 or 68 years considered), signals in December and January varied strongly and no clear trend was observed, which suggests an overlap of chilling and forcing [10].
Page 15
Point 15.1 reference lacking
Response 15.1: The Reference has been added (Luedeling et al., 2013), l.511.
Point 15.2 This may be partly due to the fact that PLSR, as used in this manuscript, does not exactly determine the dormancy brake day but the beginning of the forcing phase. Authors should
elaborate more on this aspect.
Response 15.2: Please refer to response 5.1.
Point 15.3 Well, this is arguable: in the classical approach behind the dynamic model, this is exactly as it was considered: the forcing phase would not begin before the chilling phase had ended, so
the CR was forcefully fulfilled.
Response 15.3: Yes, exactly this is the rationale in both models. What we meant to express here is that the apparent mismatch in Figure 6 is likely due to the fact that this particular assumption might not be reflecting the reality, which is however better captured using the PLSR. Considering that we reformulated this section based on your comments, please refer to the response 5.1.
Point 15.4 "more often"? Iwould say "still often" reached despite rising temperatures...
Response 15.4: Thank you for your remark, we improved the formulation. What is meant here is what you pointed at in Table 3 (p.9). Since temperatures > x and
ll.524-527: This can also be because a lower boundary for effective chilling is applicable, which is more often reached with rising temperatures. In other words, warmer climates can in some cases lead to more chilling if temperatures are less frequently below the threshold to contribute to chilling.
Point 15.5 Well, it does if not used sequentially as in this manuscript. Of course, using the model nonsequentially allows to calculate chill and heat requirements from a PLSR approach, but opens a
knowledge gap when applying it for validation or for future projections: how should the overlap or gap between both phases modeled? Should a meta-model be used above CR and HR?
Response 15.5: This is a good point of discussion and we agree that especially for the use with climate projection there is a need for a better and perhaps more complex model. Especially the growing degree hour approach has to our knowledge not been much improved since it was published in 1986 (Anderson et al., 1986). With today’s understanding of biology and physiology, an improved or entirely different modelling approach should be possible. There are other labs working on it but it was out of the scope of this manuscript. Because we were interested in the flowering stage, we used the model in this sequential setup.
Point 15.6 Please elaborate: I guess you compared that with published RPIQ values for comparable conditions or to some reference(s) of general acceptance
Response 15.6: To our knowledge there is no reference of general acceptance since this is a fairly new indicator (Bellon-Maurel et al., 2010). The RPIQ is a similar to the RPD measure but not taking the whole range but the interquartile distance (between first and third quartile) as reference. RPIQ = (Q3 – Q1)/RMSE. For the RPD, a value >2 is equivalent to an R² value > 0.75 (Minasny and McBratney, 2013), which can be seen as high but it should be seen in the context of this application, where the range of the dependent variable is rather low (considering 1951-2018, the interquartile distance of ‘Conference’ is only 16 days for example). Conceptually, an RPIQ of 2 means that the number of days that the model is wrong on average (RMSE) is half the number of days of this distance, comprising half of the data points, which we considered weak.
We added the following specification:
ll.555 – 558: However, when the values were compared to the variability of the dependent variable (date of bloom), using the RPIQ the performance of both models appears weak in the majority of cases (~RPIQ < 3) since this implies the natural variability of the phenomenon is less than three times larger than the RMSE.
Point 15.7 with this stop, a reference seems to lack: why not using a comma instead?
Response 15.7: Good point, the correction has been done in l.564.
Page 16
Point 16.1 suggests
Response 16.1: The phrase has been changed as follows:
581 – 584 : The PLSR method is inherently limited to long time series but was found to be promising to assess timing of phase change between chilling and forcing. The identified dates for the beginning of forcing differ from the modelled dormancy breaks. This supports the idea that chilling and forcing processes occur simultaneously.
Point 16.2 It took me a moment to understand that Calibration stansd for RMSE of the Calibratin dataset, etc. Maybe the authors would like to improve this to avoid any confusion to other readers?
Response 16.2: Thanks for your remark. An addition has been made to the caption of the tables
ll.602-603 and ll 604 and 605: The RMSE values are given separately for the calibration and validation sets. The mean column gives the mean of both sets.
Page 18
Point 18.1 Cannot a better reference be given? Either "phyton manual", a web repository as in R... This translator must be a function/procedure... in phyton that is documented somewhere...
Response 18.1: The authors of the package do not provide an explicit reference. The URL to the repository of the used version (in the anaconda environment) has been added as follows.
663 27. Gautier, L. Rpy2, version 2.9.4; https://anaconda.org/r/rpy2, 2018
References
Anderson, J.L., Richardson, E.A., Kesner, C.D., 1986. VALIDATION OF CHILL UNIT AND FLOWER BUD PHENOLOGY MODELS FOR “MONTMORENCY” SOUR CHERRY. Acta Hortic. 71–78. https://doi.org/10.17660/ActaHortic.1986.184.7
Avermaete, T., Biely, K., Bonjean, I., Creemers, S., Larvoe, N., Lievens, E., Maes, D., Van Passel, S., Mathijs, E., 2018. SUFISA Belgium National Report.pdf (No. D.2.2). KU Leuven, U Hasselt.
Bellon-Maurel, V., Fernandez-Ahumada, E., Palagos, B., Roger, J.-M., McBratney, A., 2010. Critical review of chemometric indicators commonly used for assessing the quality of the prediction of soil attributes by NIR spectroscopy. TrAC Trends Anal. Chem. 29, 1073–1081. https://doi.org/10.1016/j.trac.2010.05.006
Chmielewski, F.-M., Blümel, K., Henniges, Y., Blanke, M., Weber, R.W.S., Zoth, M., 2011. Phenological models for the beginning of apple blossom in Germany. Meteorol. Z. 20, 487–496. https://doi.org/10.1127/0941-2948/2011/0258
Díez-Palet, I., Funes, I., Savé, R., Biel, C., de Herralde, F., Miarnau, X., Vargas, F., Àvila, G., Carbó, J., Aranda, X., 2019. Blooming under Mediterranean Climate: Estimating Cultivar-Specific Chill and Heat Requirements of Almond and Apple Trees Using a Statistical Approach. Agronomy 9, 760. https://doi.org/10.3390/agronomy9110760
Erez, A., Fishman, S., Linsley-Noakes, G.C., Allan, P., 1989. The dynamic model for rest completion in peach buds, in: II International Symposium on Computer Modelling in Fruit Research and Orchard Management 276. pp. 165–174.
Funes, I., Aranda, X., Biel, C., Carbó, J., Camps, F., Molina, A.J., Herralde, F. de, Grau, B., Savé, R., 2016. Future climate change impacts on apple flowering date in a Mediterranean subbasin. Agric. Water Manag. 164, 19–27. https://doi.org/10.1016/j.agwat.2015.06.013
Guo, L., Wang, J., Li, M., Liu, L., Xu, J., Cheng, J., Gang, C., Yu, Q., Chen, J., Peng, C., Luedeling, E., 2019. Distribution margins as natural laboratories to infer species’ flowering responses to climate warming and implications for frost risk. Agric. For. Meteorol. 268, 299–307. https://doi.org/10.1016/j.agrformet.2019.01.038
Luedeling, E., Guo, L., Dai, J., Leslie, C., Blanke, M.M., 2013. Differential responses of trees to temperature variation during the chilling and forcing phases. Agric. For. Meteorol. 181, 33–42. https://doi.org/10.1016/j.agrformet.2013.06.018
Minasny, B., McBratney, A., 2013. Why you don’t need to use RPD. Pedometron 33, 14–15.

Reviewer 3 Report
Thank you very much for the opportunity to review the manuscript "Comparing Apple and Pear Phenology and Model 2 Performance: What 7 Decades of Observations Reveal". The manuscript addresses the phenological trends for Malus domestica and Pyrus communis, in north-eastern Belgium, for the 1950-2018 period. The paper is interesting, and may be of interest to the Agronomy journal readers.
However, the paper requires a number of changes.
At the beginning, a paragraph should be added at the beginning, describing the importance of the topic of the manuscript.
In Results and Discussion the authors can explain more clearly the differences between the results of the models and specify the advantages, disadvantages and limitations of each model. Do the authors think that a new model is needed to eliminate these shortcomings?
At line 423 - is it necessary to specify (Error! Not a valid bookmark self-reference.)?
At lines 520-521 "Advances of bloom dates with warmer dormancy periods have been found to be stronger for pear than for apple" or at Results, the authors should explain clearly the causality of these differences.
Author Response
Response to reviewer 3
Dear reviewer,
We thank you for taking your time to review our manuscript and giving us the opportunity to further improve it. Please find the revised version of our article “Comparing Apple and Pear Phenology and Model Performance: What 7 Decades of Observations Reveal”. We greatly acknowledge your comments and suggestions. Below, we provide an extensive answer to these comments.
With kind regards,
Bianca Drepper and co-authors
Point 1: At the beginning, a paragraph should be added at the beginning, describing the importance of the topic of the manuscript.
Response 1: Thank you for your remark. We have added a paragraph to underline the link between the first two paragraphs of the text describing the recent situation of the local Flemish Fruit sector and the body of the manuscript as below:
l.46-49 “Therefore a better understanding is needed of the impacts of warming on phenology in the specific case of apple and pear in Flanders. By means of adapted phenological models, variations in flowering and the related frost-sensitive stage can be estimated across temporal and spatial scales which can support decision making regarding (new) orchard locations and cultivar selections.”
Point 2: In Results and Discussion the authors can explain more clearly the differences between the results of the models and specify the advantages, disadvantages and limitations of each model. Do the authors think that a new model is needed to eliminate these shortcomings?
Response 2: In response to this point, as to points raised by reviewer 2, we highlighted differences between the models throughout the manuscript, including a more explicit naming that underlines one key difference in the functioning of the models which is the way chilling is accumulated - in a dynamic, reversible way or in a continuous way. As we state in ll.57-58 no model is satisfying and in ll. 538-539 suggest better combined model setups are needed to capture the full processes leading towards flowering in order to have more certainty about modelling future projections.
Also in the result sections 3.2, 3.3. and discussion section 4.3 the differences in performance have been more strongly highlighted. The sections with strongest change are given below:
ll.455 ff. In Figure 8, the Ratio of Performance to InterQuartile distance (RPIQ) is given, where higher values are desirable [41] since it implies that the RMSE is smaller compared to the interannual variability of the bloom date. In this representation, one can see that the more complex optimization algorithms (SA and MCMC) perform less well and the continuous model led to more underperforming parametrisations (strong outliers) than the dynamic + GDH model. While the differences in terms of the RMSE between apple and pear are less than half a day for both models, in terms of RPIQ they are performing better for pear than for apples. This is due to higher inter quartile range (14-17 days for pears; 12 days for apples except for Boskoop (16 days)).
Based on the RMSE and MAE (Figure 7) as well as the RPIQ (Figure 8), the dynamic + GDH model is better performing, confirming the initial hypothesis. It also has a lower spread of errors, no strictly failed predictions (i.e. return ‘no value’) and higher agreement between the optimisation functions than the continuous model, implying it is less sensitive to the method used for calibration and more comparable across studies.
ll.527ff. In this study, two contrasting models were therefore calibrated to the Belgian context, where the dynamic + GDH model outperformed the continuous model, as hypothesised.
The dynamic chill model has the advantage that accumulated chilling can be partially undone by subsequent high temperatures, while the continuous model only differentiates between chill accumulation at effective temperatures and stagnation at ineffective temperatures. However, the dynamic model combined with the GDH accumulation in this setup does not allow to account for overlap of chilling and forcing [47], nor interaction terms between respective requirements [38], which calls for improved combined models that cover the full dormancy cycle. Another of its strong points is however, that slightly negative temperatures account for chilling, in contrast to the tested continuous model as adapted to German conditions [8]. Experiments on cherry twigs point to chilling effectiveness below 0°C [18]. The dynamic + GDH model used hourly data while the continuous model relied on daily observations alone, but [50] found that using hourly data was not improving the modelling effort by more than 0.24 days of RMSE. Computational expenses are higher for the dynamic model than for the continuous model, but for the studied extent this is not an important limiting factor. The processes behind the dynamic model are however also less transparent and flexible than the continuous model which can be reproduced from the equations given in this manuscript alone.
Point 3: At line 423 - is it necessary to specify (Error! Not a valid bookmark self-reference.)?
Response 3: The erroneous reference has been corrected in the revised manuscript.
Point 4: At lines 520-521 "Advances of bloom dates with warmer dormancy periods have been found to be stronger for pear than for apple" or at Results, the authors should explain clearly the causality of these differences
Response 4: We acknowledge that this is a very interesting and relevant point. We observe that pear generally blooms earlier than apple due to lower heat requirements. Also in Figure 4, one can observe more differences between fruits than between cultivars of a fruit, which points to distinct periods of sensitivity. However, it is unfortunately out of the scope of this manuscript to cover the physiological causality of these sensitivities and resulting difference in phenology changes . This is why we focus on the observed differences of flowering function of temperatures, as we did for example in ll.299 - 301 and adding a paragraph in the corresponding discussion section:
ll.482-484 “From the observations appeared that the phenology of pear is more strongly affected by warmer temperatures than apple, which could be related to lower heat requirements of pear, but further physiological research is needed to explain the difference in sensitivity.”
Round 2
Reviewer 3 Report
The authors made the requested changes.